# High-resolution cryo-EM structure of photosystem II reveals damage from high-dose electron beams

Koji Kato[1,5], Naoyuki Miyazaki[2,5], Tasuku Hamaguchi[3,5], Yoshiki Nakajima[1], Fusamichi Akita[1✉], Koji Yonekura[3,4✉] & Jian-Ren Shen[1✉]

Photosystem II (PSII) plays a key role in water-splitting and oxygen evolution. X-ray crystallography has revealed its atomic structure and some intermediate structures. However, these structures are in the crystalline state and its final state structure has not been solved. Here we analyzed the structure of PSII in solution at 1.95 Å resolution by single-particle cryo-electron microscopy (cryo-EM). The structure obtained is similar to the crystal structure, but a PsbY subunit was visible in the cryo-EM structure, indicating that it represents its physiological state more closely. Electron beam damage was observed at a high-dose in the regions that were easily affected by redox states, and reducing the beam dosage by reducing frames from 50 to 2 yielded a similar resolution but reduced the damage remarkably. This study will serve as a good indicator for determining damage-free cryo-EM structures of not only PSII but also all biological samples, especially redox-active metalloproteins.

[1] Research Institute for Interdisciplinary Science and Graduate School of Natural Science and Technology, Okayama University, Okayama, Japan. [2] Life Science Center for Survival Dynamics, Tsukuba Advanced Research Alliance (TARA), University of Tsukuba, Ibaraki, Japan. [3] Biostructural Mechanism Laboratory, RIKEN Spring-8 Center, Hyogo, Japan. [4] Institute of Multidisciplinary Research for Advanced Materials, Tohoku University, Aoba-ku, Sendai, Japan. [5] These authors contributed equally: Koji Kato, Naoyuki Miyazaki, Tasuku Hamaguchi. ✉email: fusamichi_a@okayama-u.ac.jp; yone@spring8.or.jp; shen@cc.okayama-u.ac.jp

Photosystem II (PSII) is a multi-subunit pigment-protein complex embedded in the thylakoid membranes of higher plants, green algae, and cyanobacteria, and is the only molecular machine capable of oxidizing water by use of visible light in the nature. Water molecules are split into electrons, hydrogen atoms and oxygen molecules at the catalytic center of PSII, namely, the oxygen-evolving complex (OEC), through four electron and proton removing steps as described by the Si-state cycle (with $i = 0–4$, where $i$ indicates the number of oxidative equivalents accumulated)[1].

In order to elucidate the mechanism of the water-splitting reaction, the structure of PSII has been studied extensively by X-ray diffraction (XRD) with synchrotron radiation (SR)[2–8] and a number of spectroscopic methods including extended X-ray absorption fine structure (EXAFS) and electron paramagnetic resonance measurements[9]. The SR structure of PSII at an atomic resolution revealed that OEC is a $Mn_4CaO_5$ cluster organized into a distorted-chair form, in which the cuboidal part is composed of $Mn_3CaO_4$ and the outer manganese is attached to the cuboid via two μ-oxo-bridges[6]. However, based on the EXAFS analysis, the dose used for collecting the SR structure at 1.9 Å resolution may cause 25% of the Mn ions in OEC to be reduced to $2^+$ ions, causing some elongations in the Mn–Mn distances in the structure[10]. This issue is overcome by the use of X-ray free-electron lasers (XFEL), which provide X-ray pulses with ultra-short durations that enable collection of the diffraction data before onset of the radiation damage (diffraction before destruction)[11]. Using XFELs, radiation damage-free structure of PSII was solved at a high resolution by an approach called fixed-target serial rotational crystallography, which uses large, multiple PSII crystals by a shot-and-move/rotational method[12,13]. The result showed a shortening of 0.1–0.2 Å in some of the Mn–Mn distances, indicating that the structure represents a structural damage-free one[13]. By a combination of serial femtosecond X-ray crystallography (SFX) with XFELs and small crystals, or the fixed-target serial rotational crystallography with cryo-trap of reaction intermediates, structures of S-state intermediates up to $S_3$-state were analyzed by pump-probe experiments where snapshot diffraction images were collected from flash-illuminated PSII crystals[14–17]. These results demonstrated the appearance of a new oxygen atom O6 (or Ox) close to O5 between Mn1 and Mn4 upon two flashes, suggesting insertion of a water molecule in the $S_2 \rightarrow S_3$ transition for O = O bond formation. However, all these studies were conducted with PSII crystals, and the efficiencies of the S-state transitions in the microcrystals were reported to be slightly lower compared with those in solution using light-induced Fourier transform infrared difference spectroscopy[18]. Moreover, it is unknown if the structure of PSII in the crystalline state is the same as those in the solution.

Cryo-electron microscopy (cryo-EM) can solve the structures of proteins in solution without crystallization, which may represent the physiological states of proteins more closely. It can also analyze the dynamic changes of proteins in solutions in the time range of ms, provided that cooling of the samples is rapid enough. In recent years, the technique of cryo-EM has been developed rapidly, and the resolutions of structures that can be solved by cryo-EM are increased dramatically[19–21]. However, there is also an issue of damage caused by the electron beam irradiation during cryo-EM data collection, even though the cryo-EM is usually conducted at a low temperature. Radiation damage has been extensively studied with X-rays, and it has been shown that the damage mainly manifests as breakage of disulfide bonds, decarboxylation of acidic amino acid residues and chemical reduction of metal centers[10,22–24]. The damage caused by electron beams has also been shown in cryo-EM analysis[20]. In order to obtain a high resolution, however, cryo-EM studies are usually

conducted at a high-dose of electron beams without paying much attention to the electron beam damage. In this paper, we analyzed the structure of PSII in ice by cryo-EM at a resolution of 1.95 Å, and investigated the electron beam damage to PSII, especially its OEC, upon dose accumulation. We show that the structure of PSII analyzed by cryo-EM may represent the physiological state more closely, as it retains the PsbY subunit. However, it suffers from a severe electron beam damage at a high-dose, and this damage was reduced at a much-decreased dose without a remarkable loss of resolution. These results not only are important for the analysis of the PSII structure in solution, but also provide important implications for all cryo-EM studies that use considerably high-doses for imaging.

## Results

**High-resolution single particle analysis of PSII.** To obtain the high-resolution structure of PSII, three datasets of single-particle images of the PSII dimer from *T. vulcanus* were collected using Thermo Fisher Scientific Titan Krios and JEOL CRYO ARM 300, respectively, at different conditions as summarized in Table 1. Because the sample for the 75 k × magnification using Titan contained 5% glycerol in the buffer, the sample was diluted ten times with the buffer without glycerol. The other samples did not contain the glycerol and were concentrated by PEG 1450 precipitation in the final step. Image processing yielded final resolutions of 2.22 Å for the dataset collected at 75 k x magnification using Titan Krios (Titan-75k), 2.20 Å for the dataset collected at 96 k × magnification using Titan Krios (Titan-96k), and 1.95 Å for the dataset collected at 60 k × magnification using CRYO ARM 300 (ARM-60k) (Tables 1 and 2, Supplementary Table 1 and Supplementary Figs. 1–4). These results indicate that the quality of the cryo-EM density maps achieved, especially with the CRYO ARM 300, were at the level comparable to those obtained with SR and XFEL previously[6,13]. The resolutions of the Titan-75k and Titan-96k data were almost the same, in spite of the different magnifications and buffers used. The resolution of the ARM-60k data was remarkably better than that of the Titan-96k data, despite that the same buffer condition was used for the two datasets. The resolution achieved by cryo-EM depends on a number of factors, including sample quality, the type and preparations of cryo-grids used, the thickness of ice in the samples, microscope alignment and imaging conditions, etc. However, the major factor affecting the resolution could be the different quality of electron beams from different instruments. The CRYO ARM 300 microscope has a cold field emission gun (CFEG) that produces an electron beam with a high temporal-coherence and superior high-resolution signals[25] over that from the Schottky emission gun equipped in the Krios microscope.

In Fig. 1, the squared inverse resolution of reconstructions achieved from random subsets of particles is plotted against the subset size on a logarithmic scale. This is known as Rosenthal-Henderson plot[26]. These plots indicated that the resolution is proportional to the log of particle number. The B-factors estimated from these plots are 60.8 Å$^2$ for the Titan-75k dataset, 74.9 Å$^2$ for the Titan-96k dataset and 43.3 Å$^2$ for the ARM-60k dataset. These values are approximately proportional to the resolution of each dataset, and the ARM-60k dataset again shows the lowest value, in agreement with its highest resolution.

**Overall structure of PSII.** An overall atomic model of PSII was built based on the highest 1.95 Å resolution density map reconstructed from the ARM-60k dataset. At this resolution, the features of cofactors and water molecules can be easily identified in the map (Fig. 2). The overall architecture of the PSII dimer from

**Table 1 Sample preparation and cryo-EM data collection parameters.**

| Dataset | Titan-75K | Titan-96K | ARM-60K |
|---|---|---|---|
| Final sample conc. | 0.38 Chl mg/ml | 1.94 Chl mg/ml | 1.94 Chl mg/ml |
| **Data collection and processing** | | | |
| Microscope | Titan Krios | Titan Krios | CRYO ARM 300 |
| Detector | Falcon3EC in EC mode | Falcon3EC in EC mode | Gatan K2 summit in Counting mode |
| Magnification | 75 K | 96 K | 60 K |
| Voltage (kV) | 300 | 300 | 300 |
| Electron dose (e$^-$Å$^{-2}$) | 40 | 40 | 83 |
| Defocus range (μm) | −1.00 to −2.00 | −1.00 to −2.50 | −0.8 to −1.6 |
| Pixcel size (Å) | 0.870 | 0.678 | 0.822 |
| Symmetry imposed | C2 | C2 | C2 |
| Exposure time (s) | 45.11 | 24.64 | 10.00 |
| Number of micrographs | 2,084 | 4,237 | 2,160 |
| Number of frames per image | 78 | 39 | 50 |
| Initial particle images | 354,233 | 612,287 | 444,729 |
| Final particle images | 90,897 | 203,912 | 174,099 |
| Map resolution (Å) | 2.23 (2.22)$^a$ | 2.26 (2.20)$^a$ | 1.98 (1.95)$^a$ |
| Map sharpening B-factor (Å$^2$) | −54 (−53)$^a$ | −60 (−56)$^a$ | −34 (−32)$^a$ |
| Rosenthal-Henderson B-factor (Å$^2$) | 60.8 | 74.9 | 43.3 |

$^a$After micelle-density subtraction.

**Table 2 Statistics of data collection, processing and refinement.**

| Refinement | High-dose | Low-dose |
|---|---|---|
| PDB ID | 7D1T | 7D1U |
| EMDB ID | EMD-30547 | EMD-30548 |
| **Data collection and processing** | | |
| Total electron dose (e$^-$Å$^{-2}$) | 83 | 3.3 |
| Number of frames per image | 50 | 2 |
| Map resolution (Å) | 1.95 | 2.08 |
| Map sharpening B-factor (Å$^2$) | −32 | −29 |
| **Refinement** | | |
| Initial model used (PDB code) | 3WU2 | 3WU2 |
| Model resolution (Å) | 1.94 | 2.03 |
| FSC threshold | 0.5 | 0.5 |
| Model composition | | |
| Non-hydrogen atoms | 41,708 | 41,680 |
| Ligands | 8,854 | 8,854 |
| Waters | 2,432 | 2,121 |
| B-factors (Å$^2$) | | |
| Protein | 20.6 | 21.2 |
| Ligand | 23.5 | 24.1 |
| Water | 26.4 | 23.7 |
| R.m.s deviations | | |
| Bond lengths (Å) | 0.012 | 0.010 |
| Bond angles (°) | 1.46 | 1.31 |
| Validation | | |
| MolProbity score | 1.3 | 1.5 |
| Clashscore | 1.8 | 2.2 |
| Poor rotamers (%) | 2.4 | 3.1 |
| EMRinger score | 7.0 | 6.8 |
| Q-Score | 0.87 | 0.86 |
| Ramachandran plot | | |
| Favored (%) | 97.93 | 97.55 |
| Allowed (%) | 1.88 | 2.26 |
| Disallowed (%) | 0.19 | 0.19 |

*T. vulcanus* is very similar to that of SR (PDB: 3WU2)[6] and XFEL structures (PDB: 4UB6 and 4UB8)[13], and all pigments found in the X-ray structure are present in the cryo-EM map (Supplementary Table 2). However, we found the PsbY subunit in the cryo-EM structure, which is present in one of the two monomers in the native (PDB: 4UB6)[13] and the Sr$^{2+}$-substituted PSII dimer structures (PDB: 4IL6)[27] but absent in the SR structure (PDB: 3WU2)[6]. The lack of one PsbY subunit in one of the two monomers in the X-ray structure is caused by the steric hindrance of the PSII unit in the crystal packing. However, even if the 3D classification and 3D reconstruction are performed without imposing the C2 symmetry, this density was still seen in both sides of the PSII dimer in the cryo-EM structure, although the density is somewhat poorer compared with that of the other assigned subunits (Fig. 2a, f, and Supplementary Fig. 5). This clearly indicates the presence of two PsbY subunits in the PSII dimer in solution, suggesting that the cryo-EM structure more closely represents the native state of PSII.

The root-mean-square deviation (RMSD) is 0.40 Å for 5227 C$_\alpha$ atoms between the structures of cryo-EM and SR, and 0.46 Å for 5267 C$_\alpha$ atoms between the structures of cryo-EM and XFEL. Because the RMSD was 0.32 Å for 5241 C$_\alpha$ atoms between the SR and XFEL structures of the PSII dimer, the cryo-EM structure is almost identical to the SR and XFEL structures at the backbone level. In the cryo-EM density map, we assigned 2432 water molecules at a contour level of 5 σ, which are slightly less than the number of those assigned in the SR and XFEL structures[6,13]. The atomic displacement parameter (ADP) of the cryo-EM structure refined with Refmac5 in reciprocal space correlated well with that of the SR structure (Supplementary Fig. 6), although it may be

somewhat overestimated in the cryo-EM structure. Since the cryo-EM map was subjected to B-factor sharpening after processing, it is not suitable to compare ADP values directly between the cryo-EM and X-ray structures. Nevertheless, the relative ADP of the atoms in the molecule appears to reflect the characteristics of the map. The average ADP for the OEC atoms (13.8 Å$^2$) was found to be lower than that observed in the overall protein atoms of the cryo-EM structure (20.6 Å$^2$), suggesting that the structure of OEC was determined more reliably than that of the overall structure. This may be due to the presence of metal ions in the Mn$_4$CaO$_5$ cluster, which gives rise to higher cryo-EM density than that of the lighter protein atoms.

**Electron beam damage and the PSII structure at reduced electron beam dosages.** Several regions of PSII were found to have different structures between cryo-EM and XFEL, which are considered to arise from electron beam induced damage. In the PsbO subunit, a disulfide bond between Cys19 and Csy41 was observed in the XFEL structure[13], but it was completely cleaved in the cryo-EM structure (Fig. 3a). In the C terminus of D1 subunit, a part of Ala344, the C-terminal residue that coordinated to the Mn$_4$CaO$_5$ cluster, flipped out from the OEC and adopted an alternative conformation in the cryo-EM structure (Fig. 3b). These are the typical sign of damage caused by the electron beam irradiation during the acquirement of cryo-EM images.

In the OEC, the positions of heavy metals were confirmed clearly and were assigned based on their highest peaks in the cryo-EM map achieved at the high-dose (Fig. 4a). In addition, five

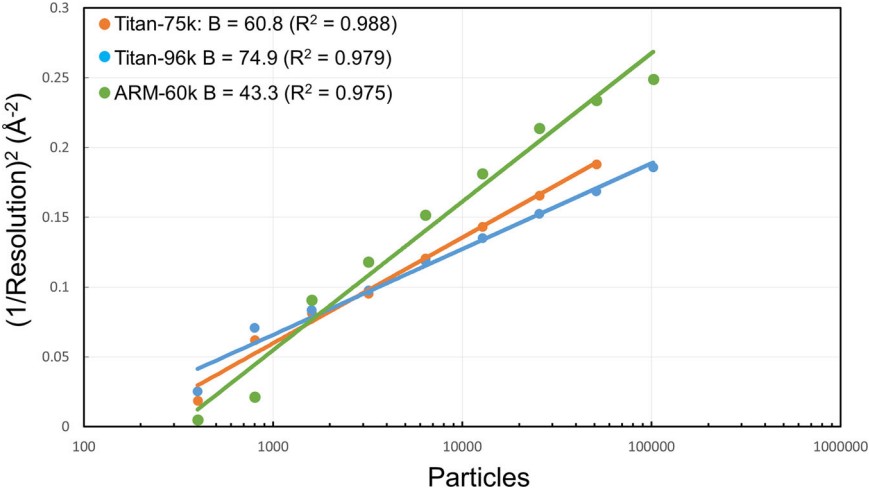

**Fig. 1 B-factor plot for the datasets of Titan-75k, Titan-96k, and ARM-60k.** B-factor plot for the Titan-75k dataset at a dose of 40 e$^{-}$Å$^{-2}$ (blue), the Titan-96k dataset at a dose of 40 e$^{-}$Å$^{-2}$ (orange), and the ARM-60k dataset at a dose of 83 e$^{-}$Å$^{-2}$ (green).

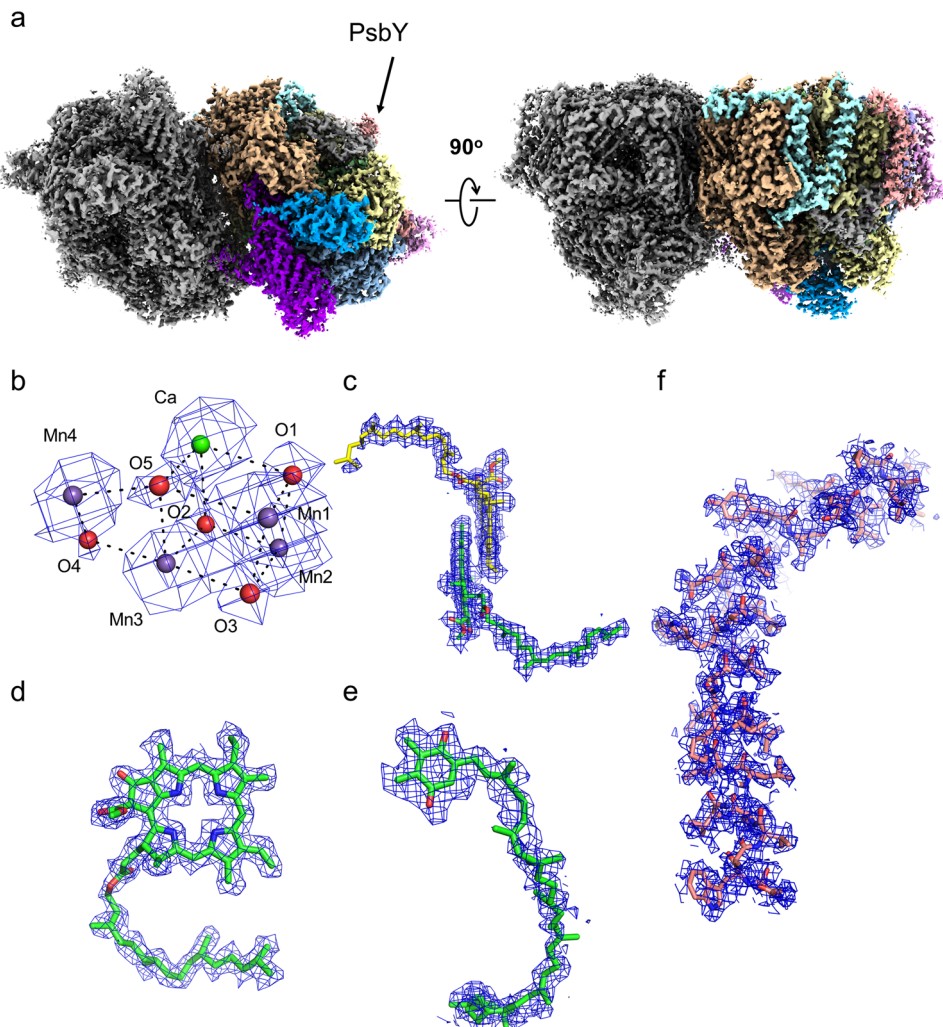

**Fig. 2 Overall structure of PSII at a high-dose. a** The cryo-EM density of PSII at 1.95 Å resolution from the ARM-60k dataset. The cryo-EM density of cofactors, OEC (**b**), P680 (**c**), pheophytin (**d**) and plastoquinone (Q$_B$) (**e**), superimposed with the refined model. **f** The density of PsbY superimposed with the refined model. The densities were depicted at 5 σ.

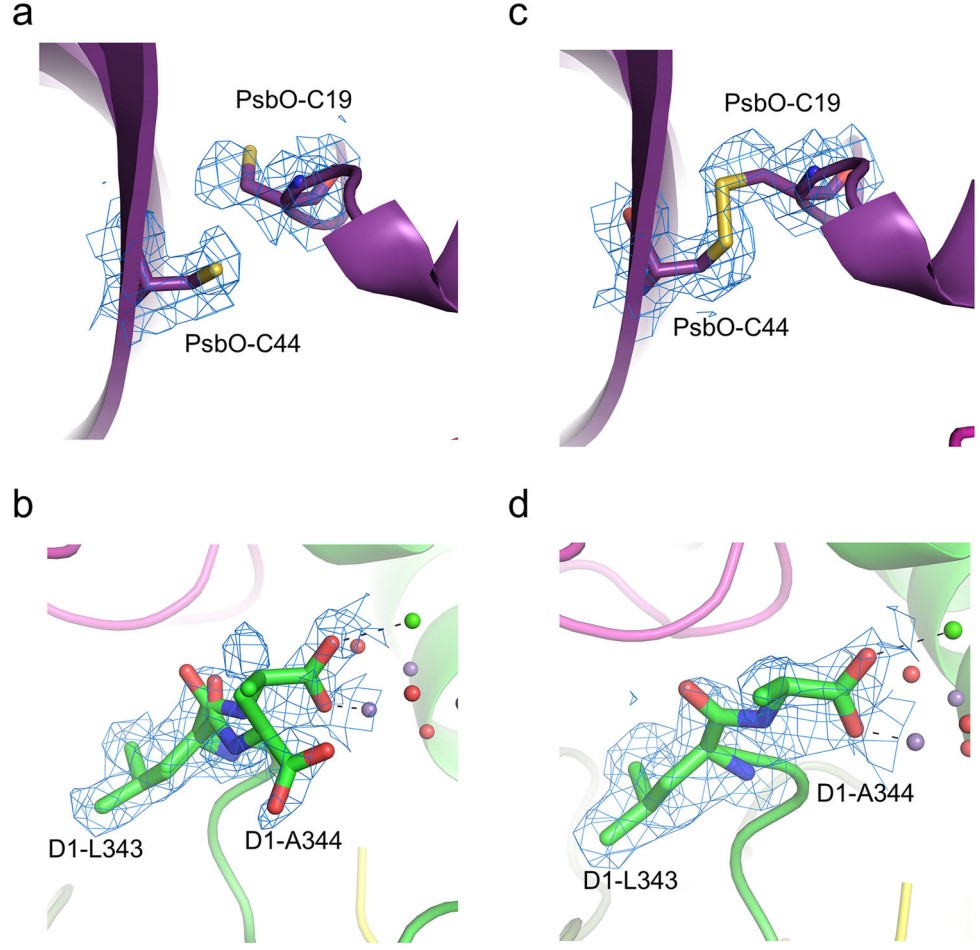

**Fig. 3 Electron beam damages in the PSII structure of the ARM-60k dataset at the high-dose (83 e⁻ Å⁻²) and low-dose (3.3 e⁻ Å⁻²). a** The broken disulfide bond in PsbO at the high-dose. **b** The alternative conformation of D1-A344 at the high dose. **c** The disulfide bond recovered in PsbO at the low-dose. **d** The single conformation of D1-A344 at the low dose. The densities were depicted at 5 σ.

oxo-oxygen atoms and four water molecules ligated to the OEC were assigned in the difference map, which were obtained by subtracting the metal densities in a diameter of 1.5 Å of that metal from the whole cryo-EM map. The overall architecture of the OEC in the cryo-EM structure is very similar to that of the SR and XFEL structures (Supplementary Fig. 7a and b); however, distinct differences were observed in Mn–Mn and Mn–O distances (Table 3). The Mn–Mn distances calculated from the initially assigned positions based on the cryo-EM density were 2.8 Å, 3.4 Å, 5.0 Å, 3.1 Å, 5.6 Å, and 2.9 Å for Mn1–Mn2, Mn1–Mn3, Mn1–Mn4, Mn2–Mn3, Mn2–Mn4, and Mn3–Mn4, respectively (Table 3). Corresponding distances in the SR structure are 2.8 Å, 3.3 Å (increased in cryo-EM structure by 0.1 Å), 5.0 Å, 2.9 Å (+0.2 Å), 5.4 Å (+0.2 Å), 3.0 Å (−0.1 Å), and corresponding distances in the XFEL structure are 2.7 Å (increased by 0.1 Å), 3.2 Å (+0.2 Å), 5.0 Å, 2.7 Å (+0.4 Å), 5.2 Å (+0.4 Å), 2.9 Å, respectively. Thus, the Mn1–Mn3, Mn2–Mn3, and Mn2–Mn4 distances of the cryo-EM structure are 0.1–0.2 Å longer than those of the SR structure[6], and except the Mn1–Mn4 and Mn3–Mn4 distances, all distances are 0.1–0.4 Å longer than those of the XFEL structure[13]. Most of the Mn–O distances in the cryo-EM structure were also 0.1–0.5 Å and 0.1–0.7 Å longer than those in the SR and XFEL structures, respectively (Table 3)[6,13]. These differences may be caused by two factors. One is the electron beam damage, and the other one is the experimental errors in determining the positions of the individual atoms based on the cryo-EM map only. Especially, the position of

oxygen atoms may not be determined precisely because the map of the oxygen atoms cannot be separated from the map of metal ions. Thus, the OEC structure was refined with the restraints for bond distances (Mn–O and Ca–O) that were taken from their initial positions. The Mn–Mn distances refined under these restraints were 2.8 Å, 3.4 Å, 5.0 Å, 3.1 Å, 5.6 Å and 3.0 Å for Mn1–Mn2, Mn1–Mn3, Mn1–Mn4, Mn2–Mn3, Mn2–Mn4, and Mn3–Mn4, respectively (Fig. 4b and Table 3). The Mn1–Mn3, Mn2–Mn3, and Mn2–Mn4 distances are 0.1–0.2 Å longer than those in the SR structure[6], and except the Mn1–Mn4 distance, all of the Mn–Mn distances are 0.1–0.4 Å longer than those of the XFEL structure[13]. Most of the Mn–O distances in the cryo-EM structure refined with the restraints were also 0.1–0.4 Å longer than those in the SR and XFEL structures (Table 3). In addition, the occupancy of the OEC atoms refined with Refmac5 was found to be lower than 1.0 (0.87). These results indicate that the OEC is reduced by electron beam exposed, leading to the elongation of the Mn–Mn and Mn–O distances and some disorder or displacement of the metal centers during the cryo-EM data acquisition. The reduced occupancy of the OEC atoms is in agreement with the previous theoretical calculation of the cryo-EM structure of higher plant PSII-LHCII supercomplex[28], in agreement with the occurrence of electron beam damage. The reduction of metal ions with electron beams was already observed in electron crystallography previously[29], and may be the reason why a part of Ala344 flipped out and does not ligate to OEC. The reduced occupancy may also represent the situation that the

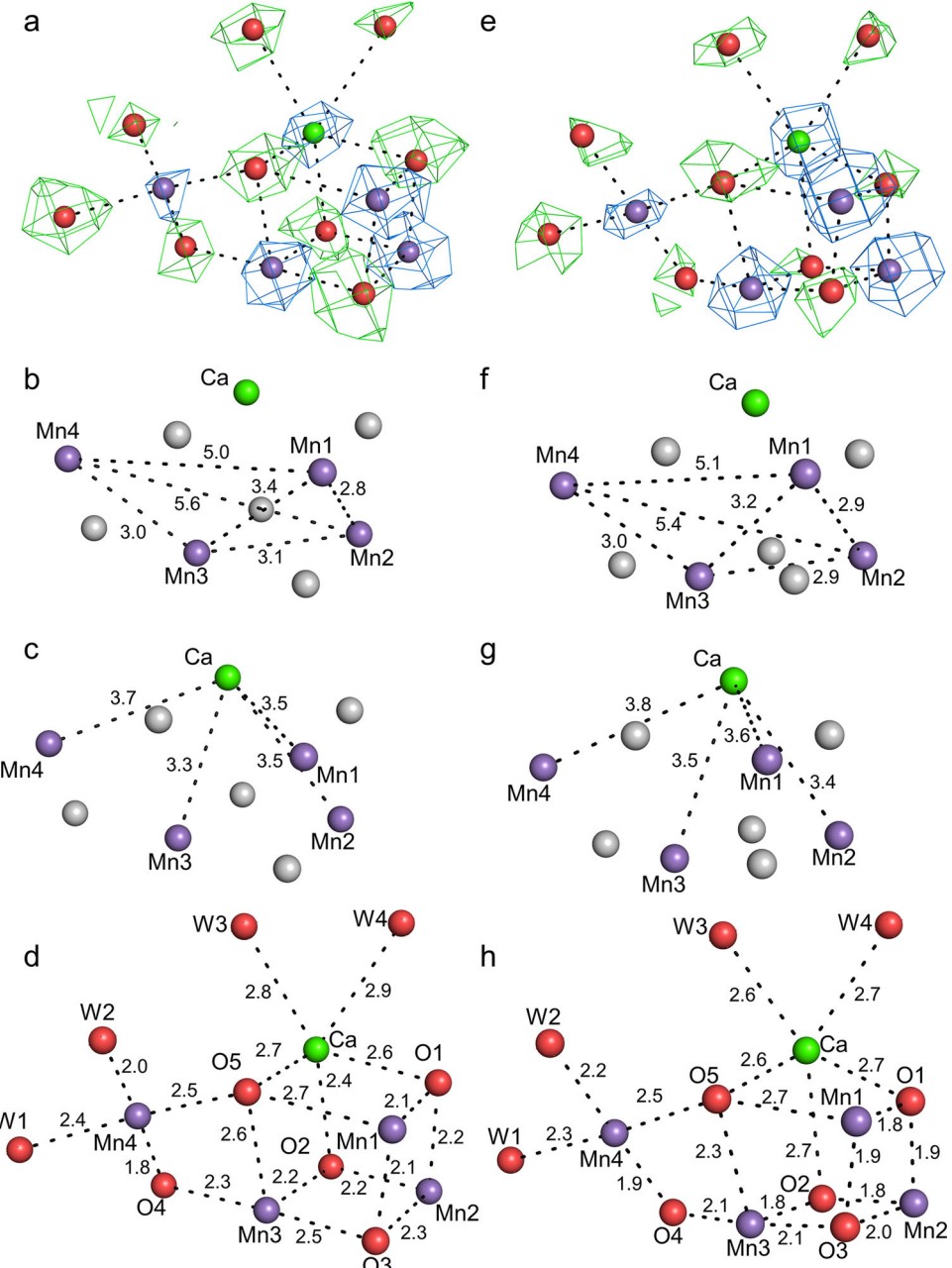

**Fig. 4 Electron beam damages in the OEC structure solved at the high-dose (83 e⁻Å⁻²) and low-dose (3.3 e⁻Å⁻²). a–d** High dose structure. **a** The cryo-EM density (blue) for manganese and calcium atoms at 17 σ and the subtracted map (green) for oxygen atoms and water molecules at 7 σ. **b** Mn–Mn distances in OEC (Å). **c** Mn–Ca distances in OEC (Å). **d** Mn–O, Ca–O, Mn-water and Ca-water distances in OEC (Å). **e–h** Low dose structure. **e** The cryo-EM density (blue) for manganese and calcium atoms at 17 σ and the subtracted map (green) for oxygen atoms and water molecules at 7 σ. **f** Mn–Mn distances in OEC (Å). **g** Mn–Ca distances in OEC (Å). **h** Mn–O, Ca–O, Mn-water, and Ca-water distances in OEC (Å).

electron form factors of metal ions are higher than those of neutral atoms[29,30].

Further structural changes in the redox-active sites, including reaction center chlorophylls, electron transfer chain, proton channels and water channels, were not found (Supplementary Fig. 7), except the water molecule near D2-Tyr160 (Y_D) (Supplementary Fig. 7c). This water molecule was disordered at two positions with one being able to hydrogen bond to Y_D and the other one being able to hydrogen bond to D2-Arg180 in the SR and XFEL structures[6,13]. In the cryo-EM structure, this water molecule was ordered and connected to D2-Arg180. This may reflect the electron beam induced damage, which causes full reduction of Y_D⁺ and breaks the hydrogen-bond to Y_D (see Methods section).

In order to reduce the electron beam damage, the final cryo-EM maps were calculated from only initial several frames of each movie stack. In Supplementary Fig. 8, the inverse resolutions of reconstructions achieved from decreased electron doses for the ARM-60k dataset, associated with the frame numbers, are plotted against the dose values on a logarithmic scale. Surprisingly, the electron doses from 83 e⁻Å⁻² down to 10 e⁻Å⁻², corresponding to the initial 1-24 frames and 1-6 frames, gave rise to almost the same resolution, indicating that increase in the electron beam dosage during this range does not contribute to increase in the resolution remarkably. Near atomic resolution is retained at the total dose of 3.3 e⁻Å⁻² for the ARM-60k dataset (2.08 Å), which was achieved by using the initial two frames of each micrograph.

**Table 3 Summarization of the distances of atoms of the $Mn_4CaO_5$ cluster.**

| | High-dose (Aver-age) | Low-dose (Aver-age) | SR (3WU2) (Aver-age) | XFEL (4UB6) (Aver-age) | High-dose | | | | | Low-dose | | | | |
|---|---|---|---|---|---|---|---|---|---|---|---|---|---|---|
| | | | | | 4th | 3rd | 2nd | 1st | Init-ial | 4th | 3rd | 2nd | 1st | Init-ial |
| Mn1–Mn2 | 2.8 | 2.9 | 2.8 | 2.7 | 2.8 | 2.8 | 2.8 | 2.8 | 2.8 | 2.9 | 2.9 | 2.9 | 2.8 | 3.0 |
| Mn1–Mn3 | 3.4 | 3.2 | 3.3 | 3.2 | 3.4 | 3.4 | 3.4 | 3.4 | 3.5 | 3.2 | 3.2 | 3.2 | 3.2 | 3.4 |
| Mn1–Mn4 | 5.0 | 5.1 | 5.0 | 5.0 | 5.0 | 5.0 | 5.0 | 5.0 | 5.0 | 5.1 | 5.1 | 5.1 | 5.0 | 5.3 |
| Mn2–Mn3 | 3.1 | 2.9 | 2.9 | 2.7 | 3.1 | 3.1 | 3.1 | 3.1 | 3.1 | 2.9 | 2.9 | 2.9 | 2.9 | 2.8 |
| Mn2–Mn4 | 5.6 | 5.4 | 5.4 | 5.2 | 5.7 | 5.7 | 5.7 | 5.6 | 5.4 | 5.5 | 5.5 | 5.4 | 5.4 | 5.4 |
| Mn3–Mn4 | 3.0 | 3.0 | 3.0 | 2.9 | 3.0 | 3.0 | 3.0 | 2.9 | 2.7 | 3.0 | 3.0 | 3.0 | 3.0 | 3.1 |
| Mn1–Ca | 3.5 | 3.6 | 3.5 | 3.5 | 3.6 | 3.5 | 3.5 | 3.5 | 3.5 | 3.6 | 3.6 | 3.6 | 3.6 | 3.8 |
| Mn2–Ca | 3.5 | 3.4 | 3.4 | 3.3 | 3.5 | 3.5 | 3.5 | 3.5 | 3.3 | 3.4 | 3.4 | 3.4 | 3.4 | 3.4 |
| Mn3–Ca | 3.3 | 3.5 | 3.4 | 3.4 | 3.3 | 3.3 | 3.3 | 3.3 | 3.0 | 3.5 | 3.5 | 3.4 | 3.5 | 3.4 |
| Mn4–Ca | 3.7 | 3.8 | 3.8 | 3.8 | 3.7 | 3.7 | 3.7 | 3.7 | 3.6 | 3.8 | 3.8 | 3.7 | 3.8 | 3.9 |
| Mn1–O1 | 2.1 | 1.8 | 1.9 | 1.8 | 2.1 | 2.1 | 2.1 | 2.1 | 2.3 | 1.8 | 1.8 | 1.8 | 1.8 | 2.1 |
| Mn1–O3 | 2.1 | 1.9 | 1.8 | 1.9 | 2.1 | 2.1 | 2.1 | 2.1 | 2.3 | 1.9 | 1.9 | 1.9 | 1.9 | 1.9 |
| Mn1–O5 | 2.7 | 2.7 | 2.6 | 2.7 | 2.7 | 2.7 | 2.7 | 2.8 | 2.6 | 2.7 | 2.7 | 2.7 | 2.7 | 2.9 |
| Mn2–O1 | 2.2 | 1.9 | 2.1 | 1.8 | 2.2 | 2.2 | 2.2 | 2.2 | 2.2 | 2.0 | 2.0 | 1.9 | 1.9 | 2.1 |
| Mn2–O2 | 2.2 | 1.8 | 2.1 | 1.8 | 2.2 | 2.2 | 2.2 | 2.2 | 2.5 | 1.8 | 1.8 | 1.8 | 1.8 | 2.5 |
| Mn2–O3 | 2.3 | 2.0 | 2.1 | 2.0 | 2.3 | 2.3 | 2.3 | 2.3 | 2.4 | 2.0 | 2.0 | 2.0 | 2.0 | 2.0 |
| Mn3–O2 | 2.2 | 1.8 | 1.9 | 1.9 | 2.2 | 2.2 | 2.2 | 2.1 | 2.0 | 1.8 | 1.8 | 1.8 | 1.9 | 2.1 |
| Mn3–O3 | 2.5 | 2.1 | 2.1 | 2.1 | 2.5 | 2.5 | 2.4 | 2.4 | 2.5 | 2.1 | 2.1 | 2.1 | 2.1 | 2.0 |
| Mn3–O4 | 2.3 | 2.1 | 2.1 | 1.9 | 2.3 | 2.3 | 2.3 | 2.3 | 2.4 | 2.2 | 2.1 | 2.1 | 2.0 | 2.7 |
| Mn3–O5 | 2.6 | 2.3 | 2.4 | 2.2 | 2.6 | 2.6 | 2.6 | 2.5 | 2.6 | 2.4 | 2.3 | 2.3 | 2.2 | 2.6 |
| Mn4–O4 | 1.8 | 1.9 | 2.1 | 2.0 | 1.8 | 1.8 | 1.8 | 1.8 | 2.0 | 1.9 | 1.9 | 1.9 | 2.0 | 2.1 |
| Mn4–O5 | 2.5 | 2.5 | 2.5 | 2.3 | 2.5 | 2.5 | 2.5 | 2.4 | 2.6 | 2.5 | 2.5 | 2.5 | 2.4 | 2.7 |
| Mn4–W1 | 2.4 | 2.3 | 2.2 | 2.3 | 2.4 | 2.4 | 2.4 | 2.4 | 2.5 | 2.3 | 2.3 | 2.4 | 2.3 | 2.3 |
| Mn4–W2 | 2.0 | 2.2 | 2.2 | 2.1 | 2.1 | 2.0 | 2.0 | 2.1 | 2.1 | 2.2 | 2.2 | 2.2 | 2.2 | 1.8 |
| Ca–O1 | 2.6 | 2.7 | 2.4 | 2.6 | 2.6 | 2.6 | 2.6 | 2.6 | 2.6 | 2.7 | 2.7 | 2.7 | 2.6 | 2.5 |
| Ca–O2 | 2.4 | 2.7 | 2.5 | 2.7 | 2.4 | 2.4 | 2.4 | 2.4 | 2.4 | 2.7 | 2.7 | 2.7 | 2.7 | 2.7 |
| Ca–O5 | 2.7 | 2.6 | 2.7 | 2.5 | 2.7 | 2.7 | 2.7 | 2.7 | 2.7 | 2.6 | 2.6 | 2.6 | 2.6 | 2.6 |
| Ca–W3 | 2.8 | 2.6 | 2.4 | 2.6 | 2.8 | 2.8 | 2.8 | 2.8 | 2.8 | 2.6 | 2.6 | 2.6 | 2.5 | 2.6 |
| Ca–W4 | 2.9 | 2.7 | 2.5 | 2.5 | 2.9 | 2.9 | 2.9 | 2.9 | 2.9 | 2.7 | 2.7 | 2.7 | 2.7 | 2.7 |

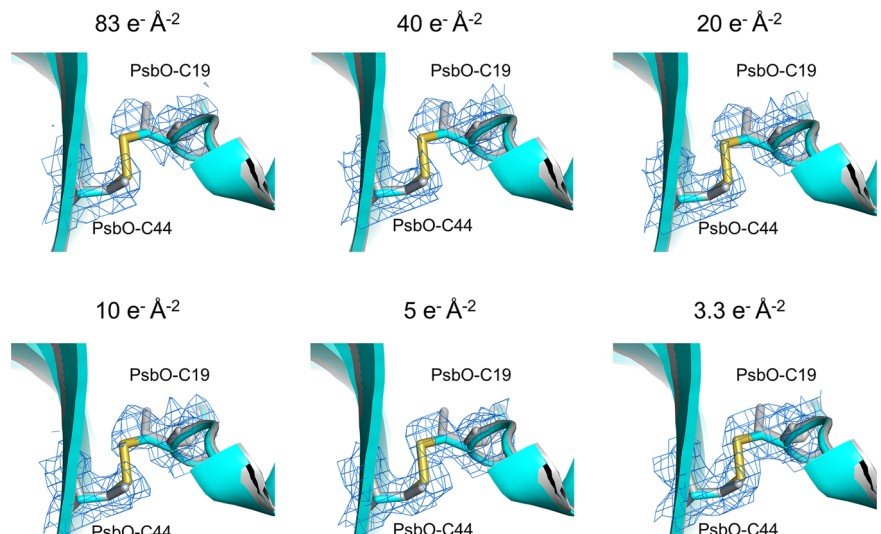

**Fig. 5 Changes of the cryo-EM map in the region of the disulfide bond in PsaO with changes of the electron beam dose.** The cryo-EM maps for each electron dose are displayed as a blue mesh at 4 σ and the corresponding models for low-dose (colored) and high-dose (gray) are shown as sticks.

An overall atomic model of the low-dose PSII was built based on the highest 2.08 Å resolution density reconstructed from the ARM-60k dataset at the dose of 3.3 e−Å−2 (Table 2). The overall architecture of the low-dose PSII is very similar to that of the high-dose PSII, with a RMSD of 0.21 Å for 5310 $C_\alpha$ atoms between the structures of high-dose and low-dose. However, in the regions where structural changes were observed due to

electron beam damage, the disulfide bond between Cys19 and Csy41 of the PsbO was retained at a dose of 5 e−Å−2, and Ala344 of the D1 subunit was returned to the single conformation to ligate the OEC similar to those seen in the crystal structures (Figs. 3–5, Supplementary Figs. 7 and 9). The ADP for the OEC atoms (12.8 Å$^2$) were lower than that observed in the overall protein atoms of the cryo-EM structure (22.4 Å$^2$), and the

occupancy value of the OEC atoms refined with Refmac5 was returned to 1.0. This occupancy is not very accurate due to utilization of the B-factor sharpening maps for the refinement of the OEC, and again this observation may represent the higher electron form factors of metal ions[29,30]. Nevertheless, when the atomic model was refined under the same conditions for the high-dose and low-dose maps, the occupancy of the OEC for the low-dose map apparently increased. These results indicate the reduction of the electron beam damage in the structure.

The Mn–Mn distances calculated from the initially assigned positions based on the low dose cryo-EM density were 3.0 Å, 3.4 Å, 5.3 Å, 2.8 Å, 5.4 Å, and 3.1 Å for Mn1–Mn2, Mn1–Mn3, Mn1–Mn4, Mn2–Mn3, Mn2–Mn4, and Mn3–Mn4, respectively (Table 3). Most of these Mn–Mn distances are 0.1–0.3 Å longer than those of the SR and XFEL structures[6,13]. Most of the Mn–O distances in the low dose cryo-EM structure were also 0.1–0.6 Å and 0.1–0.8 Å longer than those in the SR and XFEL structures, respectively (Table 3). As is done with the high-dose structure, we refined the OEC structure with the restraints for bond distances of Mn–O and Ca–O that were taken from the initial positions. The Mn–Mn distances in the OEC refined under these restraints were 2.9 Å, 3.2 Å, 5.1 Å, 2.9 Å, 5.4 Å, and 3.0 Å for Mn1–Mn2, Mn1–Mn3, Mn1–Mn4, Mn2–Mn3, Mn2–Mn4, and Mn3–Mn4, respectively (Fig. 4f, Table 3). Most of these distances are shorter than those of the high-dose structure and close to those of the XFEL structure, although most of them are still longer than the SR and XFEL structures by 0.1 Å and 0.1–0.2 Å, respectively (Table 3). The Mn–O distances after refinement under the restraints also became close to the XFEL structure, indicating the necessity of refinement with restraints. However, some of the Mn–O distances were still longer or deviated from those found in the XFEL structure, which may be caused by the electron beam damage remained and/or coordinate errors in the cryo-EM analysis at the current resolution.

## Discussion

X-ray crystal structure analysis requires crystallization of proteins, which is sometimes difficult to achieve. This is particular a problem in obtaining well-diffracting crystals of membrane proteins and their complexes. In contrast, single-particle cryo-EM can solve the structures of proteins in solution without crystallization, and the resolution of cryo-EM analysis has been improved to an atomic level recently[20,21]. Due to the improved resolution, it has been reported that biological samples are damaged by electron beams[20]. In this study, we analyzed the structure of PSII, a large membrane-protein complex that catalyzes light-induced water oxidation in photosynthesis, at 1.95 Å resolution in solution by cryo-EM. The overall structure of PSII obtained by cryo-EM analysis is very similar to the SR or XFEL structure in the crystalline state; however, the PsbY subunit was visible in the cryo-EM structure but absent or partially visible in the SR and XFEL structures[6,13,27]. This indicates that the structure solved by cryo-EM may represent the physiological state more closely.

Despite the total electron dose of 83 e⁻Å⁻² which is commonly used in the acquisition of cryo-EM images, radiation damages are found in regions susceptible to redox changes, i.e., the disulfide bond and the redox-active metals. The exposure of samples to a flux of electrons is conveniently expressed in terms of electrons per Å² of specimen surface area (e⁻Å⁻²), which is converted to the SI unit for the absorbed ionizing radiation, the Gray (Gy, with $1\text{ Gy} = 1\text{ J kg}^{-1}$), by a factor of 3.7[31]. Thus, the total electron dose of 83 e⁻Å⁻² is equal to the absorption of 307 MGy, which greatly exceeds the Henderson limit (20 MGy) that is the X-ray dose that a cryo-cooled crystal can absorb before the diffraction pattern

decays to half of its original intensity[32]. Nevertheless, our cryo-EM structure is almost the same to the SR and XFEL structures in the redox-active sites, including reaction center chlorophylls, electron transfer chain, proton channels, and water channels, indicating that the radiation damage does not affect the structure remarkably. This likely resulted from a successful dose-weighted correction in the Bayesian polishing step. However, in the PsbO subunit, a disulfide bond between Cys19 and Csy41 was completely broken (Fig. 3) in the cryo-EM structure. In the OEC structure, the Mn–Mn were 0.1–0.4 Å longer than those in the XFEL structure, and most of the Mn–O distances were also longer. In addition, the occupancy of the OEC atoms were lower than 1.0 (0.87), resulting in a multiple conformation of the C-terminal residue of D1, where a part of Ala344 flipped toward a direction that does not ligate the OEC.

We examined whether radiation damage could be reduced by reducing the total number of stacked movie frames used in the structural analysis. In the electron doses analyzed, the reconstructed map from summing the initial two frames of each micrograph (3.3 e⁻Å⁻², 12.2 MGy) gave rise to almost a similar resolution (2.08 Å) to that of the high-dose dataset (83 e⁻Å⁻², 307 MGy) (1.95 Å) (Supplementary Fig. 8). When the low-dose and high-dose cryo-EM structures were compared, it was found that a disulfide bond in PsbO was retained and A344 of the D1 subunit remained in the single conformation as found in the SR and XFEL structures (Figs. 3 and 5). This indicates a remarkable reduction of the electron beam damage. In the structure of OEC, most of the Mn–Mn and Mn–O distances became shorter than those observed in the high-dose structure before refinement. However, some of them are still remarkably deviated from those of the XFEL structure (Table 3), which become closer after the refinement with restraints taken from the initial structure. Thus, it is advisable to refine the cryo-EM structure with restraints taken from the initial positions of atoms determined based on the highest peaks of the cryo-EM map, for highly radiation-sensitive compounds such as the $Mn_4CaO_5$ cluster.

After refinement, most of the Mn–Mn and Mn–O distances of the low-dose structure are similar to those observed in the XFEL structure. However, some of the distances are still longer than or deviated from those found in the radiation damage-free XFEL structure[13]. This may be caused by three reasons. First, some electron beam damage may have remained even in the low-dose structure. Second, coordinate errors may exist in the cryo-EM structure (and XFEL structure) at this level of resolution. Finally, since the X-ray structure measures the electron density only, whereas the cryo-EM measures the electrostatic potential of nuclei and electrons together, there may be differences in the position of centers of electrons and nuclei *plus* electrons.

It has been reported that about 80% Mn ions of OEC are reduced to divalent cations at a dose of 5 MGy[7]. If this dose-reduction relationship is applied to cryo-EM, which has not yet been validated, the low dose of 12.2 MGy used in this study would reduce 90% Mn ions of the OEC to $Mn^{2+}$ ion. However, the structure of the OEC retained an occupancy of 1.0. This suggests that the radiation damage caused by electron beam may be somewhat different from that caused by X-ray irradiation. In fact, it has been reported that the damage caused by X-ray absorption and damage caused by electric wires affect different parts of proteins[33,34]. An alternative explanation is that, it may require some time for the structural changes to occur after Mn ions are reduced, and in the initial two frames of the cryo-EM images, the Mn ions cannot move out of their binding sites after reduction due to the short time of exposure. Therefore, the electron beam dose that caused reduction of 90% Mn according to the X-ray irradiation observed by EXAFS may not result in structural

changes of the $Mn_4CaO_5$ cluster as expected. However, the longer distances observed in some of the metal-oxygen distances of OEC even in the low-dose structure indicated the existence of electron beam damage. This means that the current dose of cryo-EM analysis is still too high to determine the exact structure of redox-active metals or the structure of the intermediate state with subtle structural changes. Tanaka et al. has reported that, using SR, a dose of 0.1 MGy is necessary to achieve a structure similar to that of the XFEL structure[35]. Thus, in order to achieve a damage-free structure, the electron beam dose needs to be further reduced. Fortunately, our data indicated that the resolution depends on the number of particles used, and by using more images and particles, it will be possible to lower the electron beam dose and achieve the structure at a higher resolution.

Some discrepancy between the cryo-EM and XFEL structures may come from coordinate errors associated with the model refinement and experimental errors. In the absence of spatial resolution, distribution of coordinate errors may be different between cryo-EM and XFEL structures. Structural analysis by cryo-EM at a higher resolution should eliminate such errors, and gives rise to a more accurate structure. It is also expected that improvements in the averaging and structural analysis algorithms of the cryo-EM data may improve the accuracy of the structures at the same resolutions.

Finally, the X-ray crystallography and cryo-EM may give different positions of center of atoms, as the center of electron density and center of nuclei plus electrons may be somewhat different. This is particularly true for light atoms such as hydrogen atom[20], as it does not have electrons and is difficult to be detected by X-ray crystallography. In the present study, the atoms involved in the bond distances of OEC are all non-hydrogen atoms, and the resolution is not so high as to allow discussion of the subtle differences of distances that involving hydrogen atoms. Thus, we consider that this is not the reason to account for the differences in the bond distances observed between the X-ray crystallography and the present cryo-EM study.

In summary, we show that the electron dose commonly used in cryo-EM is damaging to protein samples. However, the damaged area was limited to redox-sensitive part. Moreover, we investigated the relationship between the electron dose applied and the resolution achieved, and showed that near-atomic resolution (2.08 Å) is retained at a total dose of 3.3 $e^-Å^{-2}$, which was achieved by using the initial two frames of each micrograph. Our results suggest that it is possible to obtain a structure with less damage and high resolution by reducing the total dose and increasing the number of particles. This study will serve as a good indicator for determining damage-less cryo-EM structures of PSII and all biological samples, especially redox-active metalloproteins.

## Methods

**Growth of cells and purification of PSII**. Cells of *Thermosynechococcus vulcanus* (*T. vulcanus*) were grown in four 10 L bottles at 50 °C. Thylakoid membrane obtained from cultured cells by the freeze-thaw and osmotic shock method was solubilized with N, N-dimethyldodecylamine N-oxide to extract the crude PSII. Pure PSII with a high oxygen-evolving activity was purified from crude PSII by anion exchange chromatography[36–38] and suspended with a buffer containing 20 mM MES-NaOH (pH 6.0), 0.04% β-dodecyl-D-maltopyranoside and 5% glycerol. For the Titan-96k and ARM-60k data collection, glycerol in the buffer was removed by polyethylene glycol (PEG) precipitation and the resultant PSII was re-suspended in a buffer containing 20 mM MES-NaOH (pH 6.0), 20 mM NaCl, 3 mM $CaCl_2$, 0.04% β-dodecyl-D-maltopyranoside. All procedures for sample purification was performed in the dark or under dim green light. Therefore, the purified PSII was fixed into the dark-stable $S_1$ state.

**Cryo-EM data collection**. For cryo-EM experiments, 3-μL aliquots of the PSII sample at each condition (shown in Table 1) were applied to Quantifoil R1.2/1.3, Mo 300 mesh or Cu 200 mesh grids. The grids were incubated for 10 s in an FEI Vitrobot Mark IV at 4 °C and 100% humidity under dim fluorescent light. The grids were immediately plunged into liquid ethane cooled by liquid nitrogen and then transferred into the Titan Krios electron microscope (Thermo Fischer Scientific) equipped with a field emission gun, a Cs corrector (CEOS GmbH), and a direct electron detection camera (Falcon 3EC, Thermo Fischer Scientific), or CRYO ARM 300 (JEOL) electron microscope equipped with a cold-field emission gun, an in-column type energy filter and a direct electron detection camera (Gatan K2 summit, Gatan Inc). These microscopes were operated at 300 kV and a nominal magnification of ×75,000 (Titan-75k), ×96,000 (Titan-96k) for Titan Krios, and ×60,000 (ARM-60k) for CRYO ARM 300. Images were recorded using the Falcon 3EC detector in electron counting mode or Gatan K2 summit detector in counting mode. Micrographs were recorded with a pixel size of 0.870 Å, 0.678 Å, and 0.822 Å at a dose rate of 0.89 $e^-Å^{-2}s^{-1}$, 1.62 $e^-Å^{-2}s^{-1}$, and 8.30 $e^-Å^{-2}s^{-1}$ for Titan-75k, Titan-96k, and ARM-60k, respectively. The nominal defocus range were −1.0 to −2.0 μm, −1.0 to −2.5 μm, and −0.8 to −1.6 μm for Titan-75k, Titan-96k, and ARM-60k, respectively. Each exposure was conducted for 45.11 s, 26.64 s, and 10.00 s, and were dose-fractionated into 78, 39, and 50 movie frames for Titan-75k, Titan-96k, and ARM-60k, respectively. We acquired 2084, 4237, and 2160 images for the datasets of Titan-75k, Titan-96k, and ARM-60k, respectively.

**Cryo-EM image processing**. Movie frames were aligned and summed using the MotionCor2 software[39] to obtain a final dose weighted image. Estimation of the contrast transfer function (CTF) was performed using the CTFFIND4 program[40]. All of the following processes were performed using RELION3.0[41]. For structural analyses of the Titan-75k dataset, 354,233 particles were automatically picked from 2084 micrographs by using good classes (1345 particles) as references, which were manually picked and subjected to reference-free 2D classification. These particles were used for reference-free 2D classification. Then, 309,028 particles were selected from the good 2D classes and subjected to 3D classification with a C2 symmetry. The 1.9 Å PSII structure from *T. vulcanus* (PDB: 3WU2)[6] was employed for the initial model for the first 3D classification with 60-Å low-pass filter. As shown in the Supplementary Figs. 1 and 2, the PSII structure was reconstructed from 90,897 particles at an overall resolution of 2.22 Å. For structural analyses of the Titan-96k dataset, 612,287 particles were automatically picked from 4237 micrographs by using good classes (2131 particles) as references, which were manually picked and subjected to reference-free 2D classification. These particles were used for reference-free 2D classification. Then, 566,145 particles were selected from the good 2D classes and subjected to 3D classification with a C2 symmetry. The 2.22-Å map from Titan-75k data was employed for the initial model for the first 3D classification with a 60-Å low-pass filter. As shown in the Supplementary Figs. 1 and 2, the PSII structure was reconstructed from 203,912 particles at an overall resolution of 2.20 Å. For structural analyses of the ARM-60k dataset, 481,946 particles were automatically picked from 2160 micrographs by using good classes (1405 particles) as references, which were manually picked and subjected to reference-free 2D classification. These particles were used for reference-free 2D classification. Then, 481,927 particles were selected from the good 2D classes and subjected to 3D classification with a C2 symmetry. The 2.22-Å map from Titan-75k data was employed for the initial model for the first 3D classification with a 60-Å low-pass filter. The PSII structure was reconstructed from 174,099 particles at an overall resolution of 1.95 Å (Supplementary Figs. 3 and 4). For the low-dose maps, the summing number of movie frames were decreased in the final step of Bayesian polishing and used for reconstruction without refinement of particle positions and orientations, using RELION[41] with the command line option relion_reconstruct, and then post-processed in RELION[41]. In the ARM-60k dataset, the total doses of 83, 40, 20, 10, 5, and 3.3 $e^-Å^{-2}$ correspond to the frames of 1–50, 1–24, 1–12, 1–6, 1–3, and 1–2, respectively. All of the resolution was estimated by the gold-standard Fourier shell correlation (FSC) curve with a cut-off value of 0.143 (Supplementary Figs. 2 and 4)[42]. The local resolution was estimated using RELION[41].

**B-factor estimation**. For the B-factor plot, the total set of all particles from the final refinement was randomly resampled into smaller subsets. These subsets were subjected to 3D auto-refinement and the resulting orientations were used to calculate reconstructions for each of the two random halves used in the auto-refinement. The squared values of the resulted, estimated resolutions were then plotted against the natural logarithm of the number of particles in the subset, and B-factors were calculated from the slope of the straight line best fitted with the points in the plot (Fig. 1).

**Model building and refinement**. The 1.95-Å and 2.08-Å cryo-EM maps were used for model building of the high-dose and low-dose PSII structures, respectively. First, the crystal structure of *T. vulcanus* PSII (PDB: 3WU2) was manually fitted into each cryo-EM map using UCSF Chimera[43], and then the structures were inspected and adjusted individually with COOT[44]. The structures of high-dose PSII and low-dose PSII were then refined with phenix.real_space_refine[45] and Refmac5[46] with geometric restraints for the protein–cofactor coordination. Refmac5 was used for the refinement of ADP for all atomic models and for the occupancy of OEC with the electron form factor of neutral atoms and the normal library of OEC (OEX.cif in CCP4) in reciprocal space. The positions of four manganese atoms and one calcium atom were clearly visible in the cryo-EM map

(Figs. 2 and 4). The positions of the five oxo-oxygen atoms and four water molecules ligated to the OEC were less clear, and they were identified by the difference map in which, the maps of metal ions with a diameter of 1.5 Å from that metal ion were subtracted from the whole cryo-EM map, after placement of the manganese and calcium atoms (Figs. 2 and 4). The initial positions of metal and oxygen atoms were assigned based on the highest peaks in the cryo-EM maps. This was taken as the initial structure. Subsequently, we performed the structural refinement with loose restraints (0.1 σ) for bond distances (Mn–O and Ca–O) that were taken from the initial position. Then the refinement was performed with tighter restraints (0.05 σ) for bond distances successively using the modified new library for the bond distances. This geometry optimization procedure was repeated several times until the bond distances converged. However, the distance of Mn4–O4 in the high-dose structure, and the distances of Mn1–O1, and Mn3–O2 in the low-dose structure, were fixed to 1.8 Å, because these distances were too close and could not be refined. The averages of the distances of Mn–Mn, Mn–Ca, Mn–O, and Mn–ligand were calculated from the final four refinement steps of each PSII dimer and listed in Table 3. The final models were further validated with Q-score[47], MolProbity[48] and EMringer[49]. The statistics for all data collection and structure refinement are summarized in Tables 1 and 2. All structural figures are made by Pymol[50] or UCSF ChimeraX[51].

**Difference map analysis between low-dose PSII and high-dose PSII.** A difference map was calculated by subtracting high-dose map from the low-dose map, i.e., (low-dose PSII) minus (high-dose PSII). The rotational and translational matrix was calculated based on the refined atomic coordinates using lsqkab in CCP4[52]. A map of low-dose PSII was superposed with a map of high-dose PSII which were applied by a low-pass filter and adjusted to 2.08 Å resolution, with calculated rotational and translational matrix using maprot in CCP4[53]. The high-dose PSII map and the low-dose PSII map was normalized based on the ratio of the root mean square map density value and then the difference map was calculated using UCSF Chimera[43] with the command line option vop subtract (Supplementary Fig. 9).

**Statistics and reproducibility.** The cryo-EM data was collected from a number of grids. Individual images with bad ice were excluded from the data set by visual inspection. Data collection, processing and refinement statistics were summarized in Tables 1 and 2. The data in Fig. 1 was fitted with a linear correlation function.

**Reporting summary.** Further information on research design is available in the Nature Research Reporting Summary linked to this article.

## Data availability
Atomic coordinates and cyro-EM maps for the reported structure of PSII determined from the high-dose dataset and low-dose dataset of ARM-60k were deposited in the Protein Data Bank under accession codes 7D1T and 7D1U, respectively, and in the Electron Microscopy Data Bank under accession codes EMD-30547 and EMD-30548, respectively. The cryo-EM maps of the Titan-75k dataset and Titan-96k dataset were deposited in the Electron Microscopy Data Bank under the accession codes EMD-30549 and EMD-30550, respectively. The raw movies of the highest resolution dataset were deposited in EMPIAR under the accession code of EMPIAR-10556. Source data underlying plots shown in figures are provided in Supplementary Data 1. All relevant data are available from the authors upon request.

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

## Acknowledgements

We thank Keisuke Kawakami for the critical reading of the manuscript. This work was supported by JSPS KAKENHI No. JP20H02914 (K.K.), JP19K22396, JP20H03194 (F.A.), JP20H05087 (N.M.), JP17H06434 (J.-R.S.), JST PRESTO No. JPMJPR16P1 (F.A.), the Platform Project for Supporting Drug Discovery and Life Science Research (Basis for Supporting Innovative Drug Discovery, Life Science Research (BINDS)) of AMED No. JP18am0101072j002 (N.M.), and the Cyclic Innovation for Clinical Empowerment (CiCLE) from AMED and the JST-Mirai Program Grant Number JPMJMI20G5 (K.Y.).

## Author contributions

J.-R.S. and K.Y. conceived the project; Y.N. and F.A. purified the PSII; N.M. and T.H. collected cryo-EM images; N.M., K.K. and T.H. processed the EM data. K.K. built the structure model and refined the final models; K.K. analyzed the structure; and K.K., T.H., N.M., K.Y. and J.-R.S. wrote the paper, and all of the authors joined the discussion of the results.

## Competing interests

The authors declare no competing interests.
