## [Peer Review File · Communications Biology]

Reviewers' comments:

Reviewer #1 (Remarks to the Author):

Referee comments to Kato et al.

The paper of Kato et al. is a fantastic proof of the current status of cryo-EM capabilities on the sophisticated membrane protein photosystem II. A 1.95 Å resolution structure with cryoEM is even with a globular protein not always easy to achieve but here with a 210 kDa membrane protein (420 kDa for its biological dimer) is remarkable. As structures for X-ray synchrotron radiation (SR) and X-ray free electron laser fixed target (here just as XFEL abbreviated) from the same group are for some time published a in detail comparative analyses of the cryo-EM method has been performed.

In short no structural or functional biological findings are reported which added knowledge to the current status of the photosystem II. However, this study is worthwhile to be published in *Communications in Biology* as a guideline for future cryo-EM results in respect of validity of findings eg for the usage of electron dosage.

This study used two electron dosages with 11 MGy (low dosage) and 307 MGy (high dosage) for obtaining the cryo-EM datasets of similar resolution. These two values which are below and quite above the Henderson limit of 20 MGy. However to compare radiation damage in structures for the three methods (SR, XFEL, cryo-EM) just by dosage alone is a yardstick as the way of radiation damage is occurring has much to do with the timescale it is applied. In this respect the time resolution of the used camera for the different cryo-EM experiments should be also clearly (in an extra table not as in table 1) shown as for the SR data and the XFEL experiment. The claim of XFEL as radiation damage free must be clarified to structural damage free, as with the current XFELs the pulse length of 10-100 fs may still allow reduction of metals to a certain degree. May the next generation XFELs with fs and shorter pulses can solve this question as even the cited photosystem II XFEL data had Mn- O bond length longer than theoretically expected.

With the high dosage of 307 MGy a secondary radiation damage for a disulfide bond between cysteines is well documented (figure 5).

It would be interesting to see how the heme c coordination to Cys V 37 and V 40 is changed in comparison.

Overall this is a compelling study.

Minor points

The claim (lines 137-140) that the cryo-EM structure is more native than the crystal structures (SR, XFEL) can not be made on the fact a part of PsbY can be resolved.

Simple proteins in a crystal have to be repetitive ordered in the unit cell to contribute to the diffraction. That this peripheral part of the protein can not be resolved in the crystal structures does not mean this is not in a native state. Later the authors argue PsbY can be seen in a photosystem II state with Sr cocrystallized. This may indicate a salt is needed to stabilize that part of photosystem II to be seen in a crystal structure.

Reviewer #2 (Remarks to the Author):

Kato et al. report "High-resolution cryo-EM structure of photosystem II: effects of electron beam

damage". It is a well-conceived study and presents convincing evidence for overall results obtained. However, this manuscript requires some major revisions before being accepted for publication. A surprising finding from their careful analysis is that the resolution of a low-dose structure derived from only 2 frames of movie stack is high as that of 50 frames, which information should be included in the summary.

Major concerns are as follows and require proper revisions.

1. Lines 318-320: "Even though it is estimated that about 90% of Mn of OEC is Mn(II) at an electron beam dose of 11.1 MGy used for the low-dose structure, the structure of the OEC retained an occupancy of 1.0."

This statement is incorrect and is not consistent with known literature and should be revised properly. There is no ambiguity that the OEC with four Mn(II) ions cannot exist inside the OEC-binding site even though there is an ongoing debate on the mean oxidation state of the OEC on whether it is in the (+3,+4,+4,+3) for the four Mn centers or in the (+3,+3,+3,+3) in the S1 state. The majority of investigators prefer the assignment of the (3443) oxidation state. When all four Mn centers are reduced to Mn(II) by X-ray radiation, they no longer remain inside the OEC-binding site but are free in solution. Main signals for Mn(II) detected by EXAFS are those of free Mn(II) ion in solution after being released from the OEC-binding pocket. It is impossible to have the full occupancy of the OEC with Mn centers at +2 state. In fact, photo-assembly of the OEC is step-wise oxidation Mn(II) to Mn(III) and to Mn(IV) by adding one Mn ion to the OEC per flash of illumination. A major problem with extrapolation given by these authors is an invalid assumption that radiation-induced damage is solely proportional to radiation energy absorbed by the sample. In fact, X-ray radiation-induced protein damage differs from electron-induced damage in chemistry. X-ray radiation-induced damage to photosystem II includes oxidation of many protein residues while electron-induced damage to proteasome is mainly chemical reduction of protein residues (Wang, 2016, *Protein Sci.* 25:1407-1419; Wang et al., 2018, *Protein Sci.* 2051-2061).

2. Another concern is the reliability of estimated occupancy of the OEC. Authors should explain what kind of form factors were used in model refinement. When electron scattering factors of neutral atoms were used, there could be a large error between neutral Mn atom and Mn(IV) ion, and between neutral O atom and O²⁻ anion. This will greatly affect the accuracy of occupancy estimated for the OEC. Authors should carefully address this issue.

3. Third concern is the reliability of standard deviation-scaled difference cryo-EM maps for identification of changing structural features. Recently, Kato et al. applied the difference maps of this kind to locate chlorophyll f molecules in far-red acclimated photosystem I cryo-EM maps (Ref. 47). However, Gisriel et al. (2020, *Comm. Biology*, 3:408) challenged the validity of this method because different resolution and different overall B-factor of the two maps can introduce spurious peaks that have nothing to do with an extra formyl group. In fact, more than half of chlorophyll f molecules identified using that method do not make any chemical sense, i.e., they are likely to be incorrect. Before two aligned cryo-EM maps are used for calculation of difference maps, B-factor differences between them must be first removed by reciprocal space scaling (Wang et al., 2018, *Protein Sci.* 2051-2061). In fact, that paper should be properly cited in this study because its authors have carefully addressed the same problem of electron radiation-induced damage to protein structures in a different protein system. Structure factors inverted from post-refinement of cryo-EM maps could not be properly scaled because both included a different kind of figure-of-merit weighted factors to modulate the cryo-EM maps. Only structure factors of unsharpened cryo-EM maps can be properly scaled. For this argument, I strongly recommend that authors deposit the

unsharpened two-half maps.

Minor points

1. It is unclear how many frames of movies were used in data processing. They should be explicitly stated in the Methods section. My guess is as follows: 83, 40, 20, 10, 5, 3.3 e-/Å² corresponding to 50, 24, 12, 6, 3, and 2 frames. It should be stated whether the first image was discarded for data processing due to significant initial electron-induced motion. Given the fact that authors have already processed maps using different electron dose, a plot of FSC versus of reciprocal resolution squares as a function of electron dose should be included as a supporting figure.
2. Line 27: Delete "the" in the front "the low efficiencies"
3. Line 78: "also the issue of" should be changed to "also an issue of"
4. Line 82: "photoreduction" should be changed to "chemical reduction"
5. Line 90: "cro-EM" should be "cryo-EM"
6. Lines 103, 108, 109, 110: "x k" should be changed "k x"
7. Line 126: "the log of particle size" should be changed to "the log of particle number"
8. Lines 136, 138: "The overall" should be changed to "An overall"
9. Line 147: "The root mean square deviation" should be changed to "The root-mean-squares deviation"
10. Line 153: "waters" should be changed to "those"
11. Line 290: "breaking" should be changed to "broken"
12. Line 519: Delete the space between "radi-" and "ation"
13. Lines 592, 593: "superposed" should be changed to "superimposed"
14. Throughout the text, delete the space between "e-" and "Å-2" in the unit as well as in other units

Reviewer #3 (Remarks to the Author):

In their paper Shen and coworkers present a single particle cryo-EM structure of photosystem (PS) II from the cyanobacterium *Thermosynechococcus* (T.) *vulcanus*. The authors used three different EM machines and compared and discussed the results. They show that a resolution down to 1.95 Å can be obtained with the bestEM set-up (CRYO AMR 300). The resolution is comparable to that of the best crystal structures obtained by SR/XRD and XFEL techniques (Umena et al. 2011 and Suga et al., 2015). The comparison of cryo-EM and SR/XFEL indeed shows a high similarity of the structures – except for a few points discussed below. Of central importance in the paper is the question of the structural integrity of the oxygen evolving complex (OEC), a Mn₄Ca O₅ cluster, which is discussed in detail. The detection of the labile PsbY subunit in cryo-EM, which is lost in the crystal structures, prompted the authors to claim that the cryo-EM preparation represents the physiological state of PS II better than the crystal structures. This in itself is an interesting finding. It is also stressed that for cryo-EM experiments single crystals are not necessary which are notoriously difficult to obtain from integral membrane proteins like the photosystems.

The central issue of the present work is, however, radiation damage. Indeed, such damage is observed in the cryo-EM structure at the high doses (electron fluxes) typically used in such experiments – even at cryogenic temperatures. Radiation damage has been discussed extensively in

SR-XRD studies, also for PS II. It has been shown that XFEL is a technique to circumvent this problem to a large extent -even at room temperature. In the present paper radiation damage of PS II is studied by varying the dose of radiation in the cryo-EM experiment. This approach clearly led to a significant decrease (or change) of the observed damage (-S-S- cleavage, flipping of an OEC ligand, tyrosine YD reduction/water disorder, Mn ion reduction and partial Mn ion losses). However, a low dose is also leading to a loss of resolution. Thus, the authors had to carefully optimize these two factors; this important point is nicely described in their paper. The manuscript of Kato et al. is very important for several reasons: -it shows for the first time that a very good resolution can be obtained for PSII using cryo-EM with no need for single crystals; -it demonstrates the integrity of the structure even at fairly high fluxes of electrons (with the exception of the OEC, see below); -it shows details of radiation damage of PS II both for the OEC and surrounding amino acids using cryo-EM; -the cryo-EM detects an important subunit of PS II (PsbY) that has been lost in studies using X ray crystallography; -it compares the effect of different electron beam intensities on the PS II structure; and -it compares the performance of different cryo-EM set-ups and gives possible underlying reasons for the differences observed. I think that the experiments have been well planned and performed and that the data obtained are technically sound. The main results (high resolution cryo-EM) on PS II are clearly novel – and so is the respective investigation of the radiation damage. The paper is well written and highly informative for a wide readership of biologists, chemists and biophysicists interested in the determination of biological structures. It should therefore be published in the journal after a revision, see below.

I have a few problems with this manuscript. My comments and questions are outlined below. These should be answered before publication can be recommended.

i) The first and most important question relates to the PS II samples. It is not clear in which state the samples (or particles) are, i.e. in which S-state the OEC is. This is of utmost importance for the discussion of the OEC structure. Since the authors compare the data with their structure obtained earlier, see Umena et al. (Nature, 2011) and Suga et al. (Nature, 2015) (page 7ff) I assume that it is in the (dark) S1 state (2 Mn³⁺ 2Mn⁴⁺). However, the structural changes observed in the cryo-EM structure at high dose - and even at low dose (Table 3) - suggest that a very significant reduction of the Mn ions (to Mn²⁺) took place in the beam during data collection leading to larger distances and possibly to (partial) disintegration of the manganese cluster. This could pose a serious general problem to using cryo-EM for a detailed structure determination of the OEC in its different catalytic S-states – including water binding and conversion. The authors should comment on these points and discuss the suitability and limits of cryo-EM in this and related cases. This would be very valuable for the reader.

ii) Another point to discuss is the temperature, i.e. the freezing of the samples for cryo-EM. Can such a structure be compared with room temperature data delivered by XFEL? Which effects of the freezing procedure are expected? Certainly, cryo-EM is not suitable for getting structures at higher temperatures. These limiting points could also be discussed by the authors.

iii) The authors describe the structural refinement of PS II and the OEC in particular for the cryo-EM data. Deviations of the Mn-Mn (up to 0.4 Å) and Mn-O distances (up to 0.6/0.8 Å) from the crystal structure (SR and XFEL) data are still quite large – even at low electron fluxes. A satisfying agreement can, however, be obtained by restraints of some bond distances (pages 12, 20). Then the question is how easy it is to obtain a good structure from cryo-EM (e.g. for redox-sensitive proteins) without having data available from independent crystal structures. This seems to be another limitation for

similar applications to novel membrane proteins. The authors should comment on this point in their paper.

iv) Although the manuscript deals with structure determination using cryo-EM (and SR, XFEL) techniques there have been many other methods used to get information about PS II. In particular, the status of the OEC cannot be determined without these spectroscopic techniques, see point(i). It would therefore be appropriate to mention and cite the respective references in the present work (e.g. recent reviews).

v) Although the manuscript is in general very well done, the figures should be improved. Fig. 1: increase thickness of lines; Fig. 2: enlarge structures, maybe use stereo figures; Fig. 4: use thicker lines, enlarge (numbers are too small); Supplementary Fig. 2, 4a,b and 6: use thicker lines and colors that could be better distinguished. In the important Fig. 8 the green mesh is hard to see.

Reviewer #4 (Remarks to the Author):

This is an interesting and important paper that explores the effects of electron beam damage on the structure of photosystem II. The work is well done and the structure of high quality. There are some unsupported statements and somewhat faulty comparisons but none are essential to the main message of the paper regarding radiation damage so can be easily revised or deleted. I recommend publication after some revisions as described below, none of which require additional experiments.

Major

1. Comparison between the 3 cryoEM datasets (page 6): Inconclusive and potentially misleading, should be removed. Since only the highest resolution dataset is discussed regarding radiation damage, the other two do not need to be included in the main text (or at all). Also the fits in Fig 1 are incorrect because the data is not linear over the fit range, yield

2. PsbY density (page 7): The authors claim PsbY was found on both PSII in the dimer, but the density is poorer. The authors need to show that this is the case **without** the application of C2 symmetry if they want to keep this statement. To support the statement that the cryoEM structure more closely represents the native state, the authors can for example perform refinement with symmetry expansion to estimate the fraction of PSII monomers containing PsbY. Otherwise, the authors should remove this statement.

3. Mn-O distances (page 9): The authors point out that the Mn-O distances in the cryoEM map are consistently longer than those in the maps determined by X-ray crystallography, and list two possible reasons. A third possible reason is the difference between the structure factors measured by X-ray crystallography, which correspond to the electron density only, vs those measured by EM, which correspond to the electrostatic potential due to nuclei and electrons together. See for example discussion regarding C-H bond lengths in Nakane et al 2020 bioRxiv. The authors need to point this out. The authors need to consider whether or not this effect affects their bond length measurements and conclusions regarding radiation damage.

4. The authors need to show some other bond distance deviations e.g. some polar bonds in the protein. The authors also need to show how was the pixel size calibrated to be 0.822 Angstrom/pixel, to eliminate that the deviations are not due to an overall scaling error.

5. Resolution vs beam damage (page 11, Supplementary Fig. 7, and Methods page 19):

From the methods text it seems like the reconstructions for Supplementary Figure 7 were performed with the indicated cumulative dose, for example reconstruction (6) includes all frames 1 - 6. Please clarify whether this is the case or not. Another map to look at the damaged structure *only* can be produced for example by summing all the frames from 6 onwards (without the early undamaged frames).

Add another panel to Supplementary Figure 7 which should show the per-frame B factor vs dose and the horizontal axis should match the horizontal axis of the resolution vs dose plot.

6. Data availability: Please deposit the highest resolution dataset (raw movies) in EMPIAR. It is not necessary for publication but would be appreciated by the field.

7. Please add more detail to the comparisons btw. the SR XFEL and cryoEM structures in the figures. In particular, it is interesting to see if particular atoms follow specific trajectories during damage or if their positions are just randomised / blurred out.

Minor

8. Writing clarity needs to be improved - perhaps copy editors can assist with this.

9. Figure 1 - if the authors decide to keep this part in the paper at all: please add more points for each particle number. Estimate errors on the slopes. Discard points at low resolution (skew the fits).

10. Line 124, page 6 - 'size' should read 'number'

11. Overall structure (p.7): Please comment whether or not all pigments, as found in the X-ray structure, are present in the cryoEM map. This is not clear from the supplementary figures.

12. Line 272, page 13 - symbol before 3.7 not displayed.

13. Discrepancy between methods for cryoEM data collection (page 17) and Table 1 (page 33): Methods say Falcon 3 linear mode, table says counting mode. The flux seems consistent with linear mode. The dose rate, as stated in the text, for all 3 experiments, does not equal the total dose divided by the exposure time as stated in the table. Please correct all of these.

14. CryoEM image processing (page 17 - 18): repetitive text for the 3 datasets can be summarised in Table 1 instead.

Please state which program/method was used for particle picking.

15. Supplementary Figs 1 & 3. Please state how the micrograph contrast was adjusted, low pass filtering, etc.

Responses to the comments of Reviewer #1

Comment 1:

The paper of Kato et al. is a fantastic proof of the current status of cryo-EM capabilities on the sophisticated membrane protein photosystem II. A 1.95 Å resolution structure with cryoEM is even with a globular protein not always easy to achieve but here with a 210 kDa membrane protein (420 kDa for its biological dimer) is remarkable. As structures for X-ray synchrotron radiation (SR) and X-ray free electron laser fixed target (here just as XFEL abbreviated) from the same group are for some time published a in detail comparative analyses of the cryo-EM method has been performed.

In short no structural or functional biological findings are reported which added knowledge to the current status of the photosystem II. However, this study is worthwhile to be published in Communications in Biology as a guideline for future cryo- EM results in respect of validity of findings eg for the usage of electron dosage.

Author reply 1:

First of all, we thank the reviewer for his/her highly positive and valuable comments. According to the reviewer's comments, we modified the manuscript as follows.

Comment 2:

This study used two electron dosages with 11 MGy(low dosage) and 307 MGy (high dosage) for obtaining the cryo-EM datasets of similar resolution. These two values which are below and quite above the Henderson limit of 20 MGy. However to compare radiation damage in structures for the three methods (SR,XFEL,cryo-EM) just by dosage alone is a yardstick as the way of radiation damage is occurring has much to do with the timescale it is applied. In this respect the time resolution of the used camera for the different cryo-EM experiments should be also clearly (in an extra table not as in table 1) shown as for the SR data and the XFEL experiment. The claim of XFEL as radiation damage free must be clarified to structural damage free, as with the current XFELs the pulse length of 10-100 fs may still allow reduction of metals to a certain degree. May the next generation XFELs with fs and shorter pulses can solve this question as even the cited photosystem II XFEL data had Mn- O bond lengthlonger than theoretically expected.

With the high dosage of 307 MGy a secondary radiation damage for a disulfide bond between cysteines is well documented (figure 5).

It would be interesting to see how the heme c coordination to Cys V 37 and V 40 is changed in

comparison.

Overall this is a compelling study.

Author reply 2:

Thanks for this important comment. According to the reviewer's comment, we added the parameters of cryo-EM data collection in Supplementary table 1. We further revised the original "radiation damage free" to "structural damage free" on lines 57, 61 and 321. As shown in the figure below, there is no big difference among the structures of high-dose, low-dose and XFEL regarding the binding between CysV37, V40 and heme c. In the figure, the structures of the high-dose (cyan), low-dose (magenta) and XFEL (4UB6) (gray) were superimposed with each other.

Minor points

Comment 3:

The claim (lines 137-140) that the cryo-EM structure is more native than the crystal structures (SR, XFEL) can not be made on the fact a part of PsbY can be resolved.

Simple proteins in a crystal have to be repetitive ordered in the unit cell to contribute to the diffraction. That this peripheral part of the protein can not be resolved in the crystal structures does not mean this is not in a native state. Later the authors argue PsbY can be seen in a photosystem II state with Sr cocrystallized. This may indicate a salt is needed to stabilize that part of photosystem II to be seen in a crystal structure.

Author reply 3:

As described in the “Overall structure of PSII” section (lines 137-141), the density of PsbY is present in one of the two monomers in both the native (PDB: 4UB6) and the Sr²⁺-substituted PSII dimer structures (PDB: 4IL6). However, the other PsbY subunit has some steric hindrance with the PsbU subunit from the adjacent PSII dimer in the crystal packing (figure below), and is not related with the presence of a salt. In the figure below, the PSII monomer with PsbY (sphere model colored in magenta) is superimposed to the other monomer and crystallographically symmetric PSII is colored in gray. A red circle shows steric hindrance part. From this fact, we believe that one PsbY was dissociated from PSII in the crystal structure due to the influence of the crystal packing.

To explain this, we added a sentence “The lack of one PsbY subunit in one of the two monomers in the X-ray structure is caused by the steric hindrance of the PSII unit in the crystal packing.”, in Overall structure of PSII section (lines 139-141, page 7).

Responses to comments of Reviewer #2

Comment 1:

Kato et al. report “High-resolution cryo-EM structure of photosystem II: effects of electron beam damage”. It is a well-conceived study and presents convincing evidence for overall results obtained. However, this manuscript requires some major revisions before being accepted for publication. A surprising finding from their careful analysis is that the resolution of a low-dose structure derived from only 2 frames of movie stack is high as that of 50 frames, which information should be included in the summary.

Author reply 1:

First of all, we thank the reviewer for his/her highly positive and valuable comments. According to the reviewer’s comments, we added the sentence “and reducing the beam dosage by reducing frames from 50 to 2 yielded a similar resolution but reduced the damage significantly.” into the Abstract, and the following sentence into the Summary (p16, lines 369-371).

“Moreover, we investigated the relationship between the electron dose applied and the resolution achieved. Near atomic resolution (2.08 Å) is retained at the total dose of 3.3 e⁻Å⁻², which was achieved by using the initial 2 frames of each micrograph.”

Major concerns are as follows and require proper revisions.

Comment 2:

Lines 318-320: “Even though it is estimated that about 90% of Mn of OEC is Mn(II) at an electron beam dose of 11.1 MGy used for the low-dose structure, the structure of the OEC retained an occupancy of 1.0.”

This statement is incorrect and is not consistent with known literature and should be revised properly. There is no ambiguity that the OEC with four Mn(II) ions cannot exist inside the OEC-binding site even though there is an ongoing debate on the mean oxidation state of the OEC on whether it is in the (+3,+4,+4,+3) for the four Mn centers or in the (+3,+3,+3,+3) in the S1 state. The majority of investigators prefer the assignment of the (3443) oxidation state. When all four Mn centers are reduced to Mn(II) by X-ray radiation, they no longer remain inside the OEC-binding site but are free in solution. Main signals for Mn(II) detected by EXAFS are those of free Mn(II) ion in solution after being released from the OEC-binding pocket. It is impossible to have the full occupancy of the OEC with Mn centers at +2 state. In fact, photo-assembly of the OEC is step-wise oxidation Mn(II) to Mn(III) and to Mn(IV) by adding one Mn ion to the OEC per flash of illumination. A major problem with extrapolation given by these authors is an

invalid assumption that radiation-induced damage is solely proportional to radiation energy absorbed by the sample. In fact, X-ray radiation-induced protein damage differs from electron-induced damage in chemistry. X-ray radiation-induced damage to photosystem II includes oxidation of many protein residues while electron-induced damage to proteasome is mainly chemical reduction of protein residues (Wang, 2016, *Protein Sci.* 25:1407-1419; Wang et al., 2018, *Protein Sci.* 2051-2061).

Author reply 2:

Thanks for this important comment. According to the reviewer's comment, we replaced the sentences "Even though it is estimated that about 90% of Mn of OEC is reduced to Mn(II) at an electron beam dose of 12.2 MGy used for the low-dose structure, the structure of the OEC retained an occupancy of 1.0. This may be contributed by the stability of the structure of OEC, as the metal ions of OEC are liganded by seven amino acid residues (D1-D170, D1-E189, D1-H332, D1-E333, D1-D342, D1-A344, and CP43-E354).", to the following ones and cited the papers of the radiation-induced damage (Wang, 2016, *Protein Sci.* 25:1407-1419; Wang et al., 2018, *Protein Sci.* 27:2051-2061) (page 15-16, lines 329-340).

"According to this, around 90% of Mn of OEC is estimated to be reduced to Mn(II) at an electron beam dose of 12.2 MGy used for the low-dose structural analysis. However, the structure of the OEC retained an occupancy of 1.0. This suggests that the radiation damage caused by electron beam may be somewhat different from that caused by X-ray irradiation. In fact, it has been reported that the damage caused by X-ray absorption and damage caused by electric wires affect different parts of proteins^{32,33}. An alternative explanation is that, it may require some time for the structural changes to occur after Mn ions are reduced, and in the initial two frames of the cryo-EM images, the Mn ions cannot move out of their binding sites after reduction due to the short time of exposure. Therefore, the electron beam dose that caused reduction of 90% Mn according to the X-ray irradiation observed by EXAFS may not result in structural changes of the Mn₄CaO₅ cluster as expected."

Comment 3:

Another concern is the reliability of estimated occupancy of the OEC. Authors should explain what kind of form factors were used in model refinement. When electron scattering factors of neutral atoms were used, there could be a large error between neutral Mn atom and Mn(IV) ion, and between neutral O atom and O²⁻ anion. This will greatly affect the accuracy of occupancy estimated for the OEC. Authors should carefully address this issue.

Author reply 3:

Thanks for this important comment. We estimated the atomic displacement parameter (ADP) and occupancy by using Refmac5-ccpem with the electron diffraction form factor and the normal library for the OEC (OEX.cif in CCP4) in reciprocal space. These two parameters are closely related and both need to be refined, as you pointed out. We agree with the reviewer that the OEC occupancy is not very accurate, because the B-factor sharpening map was used for the ADP refinement as described in the text (page 8, first paragraph). However, when the atomic model was refined under the same conditions for the two maps, high-dose and low-dose, the occupancy of the refined OEC model against the low-dose map was larger. This indicates that electron radiation damage has been reduced. We added the sentence “Although this occupancy is not very accurate due to the utilization of the B-factor sharpening maps for the refinement of the OEC, when the atomic model was refined under the same conditions for the high-dose and low-dose maps, the occupancy of the OEC for the low-dose map apparently increased.” in page 11, lines 242-245. Moreover, we added the sentence “Refmac5 was used for the refinement of ADP for the all atomic models and for the occupancy of OEC with the electron diffraction form factor and the normal library for the OEC (OEX.cif in CCP4) in reciprocal space.” in the Methods section (page 21, lines 462-464).

Comment 4:

Third concern is the reliability of standard deviation-scaled difference cryo-EM maps for identification of changing structural features. Recently, Kato et al. applied the difference maps of this kind to locate chlorophyll f molecules in far-red acclimated photosystem I cryo-EM maps (Ref. 47). However, Gisriel et al. (2020, *Comm. Biology*, 3:408) challenged the validity of this method because different resolution and different overall B-factor of the two maps can introduce spurious peaks that have nothing to do an extra formyl group. In fact, more than half of chlorophyll f molecules identified using that method do not make any chemical sense, i.e., they are likely to be incorrect. Before two aligned cryo-EM maps are used for calculation of difference maps, B-factor differences between them must be first removed by reciprocal space scaling (Wang et al., 2018, *Protein Sci.* 2051-2061). In fact, that paper should be properly cited in this study because its authors have carefully addressed the same problem of electron radiation-induced damage to protein structures in a different protein system. Structure factors inverted from post-refinement of cryo-EM maps could not be properly scaled because both included a different kind of figure-of-merit weighted factors to modulate the cryo-EM maps. Only structure factors of unsharpened cryo-EM maps can be properly scaled. For this argument, I strongly recommend that authors deposit the unsharpened two-half maps.

Author reply 4:

Thanks for this important comment. In this study, the resolutions of high-dose data and low-dose data are quite high and better than 2.3 Å, which is the resolution that Gisriel et al. (2020, *Comm. Biology*, 3:408) report to distinguish between methyl and formyl groups. The B-factors calculated from high-dose data and low-dose data based on Guinier plot, which are similar to Willson plot in crystallography, are very similar (32 and 29, respectively, Table 2). The difference map was calculated with the same resolution (2.08 Å) by subjecting the high-dose map to low-pass filter as described in the Methods section. Moreover, we believe that the difference map in Supplementary figure 8 qualitatively reflects the changes in the cryo-EM map that accompany the changes in the dose amount as shown in figures 3, 4 and 5. According to the reviewer's comment, we cited the paper (Wang et al., 2018, *Protein Sci.* 2051-2061) in a suitable position and we deposited the unsharpened two-half maps in the Electron Microscopy Data Bank.

Minor points

Comment 5:

It is unclear how many frames of movies were used in data processing. They should be explicitly stated in the Methods section. My guess is as follows: 83, 40, 20, 10, 5, 3.3 e⁻/Å² corresponding to 50, 24, 12, 6, 3, and 2 frames. It should be stated whether the first image was discarded for data processing due to significant initial electron-induced motion. Given the fact that authors have already processed maps using different electron dose, a plot of FSC versus of reciprocal resolution squares as a function of electron dose should be included as a supporting figure.

Author reply 5:

We have shown the plot of resolutions achieved against the electron doses in Supplementary Fig. 8. In this figure, we also indicated the number of frames in the parentheses. According to the reviewer's comment, we added the sentence "In the ARM-60k dataset, the total doses of 83, 40, 20, 10, 5 and 3.3 e⁻/Å² correspond to the total frames of 1-50, 1-24, 1-12, 1-6, 1-3, and 1-2, respectively." in the Methods section (page 20, lines 442-443).

Comment 6:

Line 27: Delete "the" in the front "the low efficiencies"

Author reply 6:

We removed it; thank you.

Comment 7

Line 78: “also the issue of” should be changed to “also an issue of”

Author reply 7:

We revised it as suggested.

Comment 8:

Line 82: “photoreduction” should be changed to “chemical reduction”

Author reply 8:

We revised it as suggested.

Comment 9:

Line 90: “cro-EM” should be “cryo-EM”

Author reply 9

We revised it as suggested.

Comment 10:

Lines 103, 108, 109, 110: “x k” should be changed “k x”

Author reply 10:

We revised it as suggested.

Comment 11:

Line 126: “the log of particle size” should be changed to “the log of particle number”

Author reply 11:

We revised it as suggested.

Comment 12:

Lines 136, 138: “The overall” should be changed to “An overall”

Author reply 12:

We revised it as suggested.

Comment 13:

Line 147: “The root mean square deviation” should be changed to “The root-mean-squares deviation”

Author reply 13:

We revised it as suggested.

Comment 14:

Line 153: “waters” should be changed to “those”

Author reply 14

We revised it as suggested.

Comment 15

Line 290: “breaking” should be changed to “broken”

Author reply 15

We revised it as suggested.

Comment 16

Line 519: Delete the space between “radi-” and “ation”

Author reply 16

We revised it as suggested.

Comment 17

Lines 592, 593: “superposed” should be changed to “superimposed”

Author reply 17

We revised it as suggested.

Comment 18

Throughout the text, delete the space between “e-” and “Å-2” in the unit as well as in other units

Author reply 18

We revised it as suggested; thank you.

Responses to comments of Reviewer #3

Comment 1:

In their paper Shen and coworkers present a single particle cryo-EM structure of photosystem (PS) II from the cyanobacterium *Thermosynechococcus* (T.) *vulcanus*. The authors used three different EM machines and compared and discussed the results. They show that a resolution down to 1.95 Å can be obtained with the bestEM set-up (CRYO AMR 300). The resolution is comparable to that of the best crystal structures obtained by SR/XRD and XFEL techniques (Umena et al. 2011 and Suga et al., 2015). The comparison of cryo-EM and SR/XFEL indeed shows a high similarity of the structures – except for a few points discussed below. Of central importance in the paper is the question of the structural integrity of the oxygen evolving complex (OEC), a Mn₄Ca O₅ cluster, which is discussed in detail. The detection of the labile PsbY subunit in cryo-EM, which is lost in the crystal structures, prompted the authors to claim that the cryo-EM preparation represents the physiological state of PSII better than the crystal structures. This in itself is an interesting finding. It is also stressed that for cryo-EM experiments single crystals are not necessary which are notoriously difficult to obtain from integral membrane proteins like the photosystems.

The central issue of the present work is, however, radiation damage. Indeed, such damage is observed in the cryo-EM structure at the high doses (electron fluxes) typically used in such experiments – even at cryogenic temperatures. Radiation damage has been discussed extensively in SR-XRD studies, also for PS II. It has been shown that XFEL is a technique to circumvent this problem to a large extent -even at room temperature. In the present paper radiation damage of PS II is studied by varying the dose of radiation in the cryo-EM experiment. This approach clearly led to a significant decrease (or change) of the observed damage (-S-S-cleavage, flipping of an OEC ligand, tyrosine YD reduction/water disorder, Mn ion reduction and partial Mn ion losses). However, a low dose is also leading to a loss of resolution. Thus, the authors had to carefully optimize these two factors; this important point is nicely described in their paper. The manuscript of Kato et al. is very important for several reasons: -it shows for the first time that a very good resolution can be obtained for PSII using cryo-EM with no need for single crystals; -it demonstrates the integrity of the structure even at fairly high fluxes of electrons (with the exception of the OEC, see below); -it shows details of radiation damage of PS II both for the OEC and surrounding amino acids using cryo-EM; -the cryo-EM detects an important subunit of PS II (PsbY) that has been lost in studies using X ray crystallography; -it

compares the effect of different electron beam intensities on the PS II structure; and-it compares the performance of different cryo-EM set-ups and gives possible underlying reasons for the differences observed. I think that the experiments have been well planned and performed and that the data obtained are technically sound. The main results (high resolution cryo-EM) on PSS II) are clearly novel – and so is the respective investigation of the radiation damage. The paper is well written and highly informative for a wide readership of biologists, chemists and biophysicists interested in the determination of biological structures. It should therefore be published in the journal after a revision, see below.

I have a few problems with this manuscript. My comments and questions are outlined below. These should be answered before publication can be recommended.

Author reply 1:

First of all, we thank the reviewer for his/her highly positive and encouraging comments. According to the reviewer's comments, we modified the manuscript as follows.

Comment 2:

The first and most important question relates to the PS II samples. It is not clear in which state the samples (or particles) are, i.e. in which S-state the OEC is. This is of utmost importance for the discussion of the OEC structure. Since the authors compare the data with their structure obtained earlier, see Umena et al. (Nature, 2011) and Suga et al. (Nature, 2015)(page 7ff) I assume that it is in the (dark) S₁ state (2 Mn³⁺ 2Mn⁴⁺). However, the structural changes observed in the cryo-EM structure at high dose - and even at low dose (Table 3) - suggest that a very significant reduction of the Mn ions (to Mn²⁺) took place in the beam during data collection leading to larger distances and possibly to (partial)disintegration of the manganese cluster. This could pose a serious general problem to using cryo-EM for a detailed structure determination of the OEC in its different catalytic S-states – including water binding and conversion. The authors should comment on these points and discuss the suitability and limits of cryo-EM in this and related cases. This would be very valuable for the reader.

Author reply 2:

Thanks for this important comment. The sample purification was performed in the dark or under dim green light. Therefore, the purified PSII was fixed into the dark-stable S₁ state. The cryo-grids were prepared also under a dim fluorescent light in a short time. We thus consider that the state of PSII is in the S₁-state. This is consistent with the fact that the cryo-EM structure is similar to the S₁-state XFEL structure. According to the reviewer's comment, we added the sentences "The sample purification was performed in the dark or under dim green light.

Therefore, the purified PSII was fixed into the dark-stable S1 state.”, and “under dim fluorescent light” in the Methods section (page 18, lines 386-388, and 393, respectively).

As described in the Discussion section (page 16, lines 340-342), we mentioned the possibility of damage-free structure determination by using cryo-EM. In addition to it, we added the sentence “This means that the current dose of cryo-EM analysis is still too high to determine the exact structure of redox-active metals or the structure of the intermediate state with subtle structural changes.” in the page 16, lines 342-344).

Comment 3:

Another point to discuss is the temperature, i.e. the freezing of the samples for cryo-EM. Can such a structure be compared with room temperature data delivered by XFEL? Which effects of the freezing procedure are expected? Certainly, cryo-EM is not suitable for getting structures at higher temperatures. These limiting points could also be discussed by the authors.

Author reply 3:

The SR (Umena et al. Nature, 2011) and XFEL (Suga et al. Nature, 2015) structures compared in this study, were determined under the nitrogen stream at 100 K, which is the same as the temperature used for the cryo-EM analysis. Furthermore, in the structural comparison with the SFX structure (Suga et al. Nature, 2017) whose structure was determined at room temperature, it has been reported that the structure is almost the same except for some structural changes due to S-state changes in OEC induced by light illumination. Therefore, it is considered that there are no substantial structural differences between the room temperature and low temperature.

Comment 4:

The authors describe the structural refinement of PS II and the OEC in particular for the cryo-EM data. Deviations of the Mn-Mn (up to 0.4 Å) and Mn-O distances (up to 0.6/0.8 Å) from the crystal structure (SR and XFEL) data are still quite large – even at low electron fluxes. A satisfying agreement can, however, be obtained by restraints of some bond distances (pages 12, 20). Then the question is how easy it is to obtain a good structure from cryo-EM (e.g. for redox-sensitive proteins) without having data available from independent crystal structures. This seems to be another limitation for similar applications to novel membrane proteins. The authors should comment on this point in their paper.

Author reply 4:

Thanks for this important comment. In this study, we used the high resolution SR structure (3WU2, Umena et al. Nature, 2011) as the initial atomic model for the refinement. However, the

cryo-EM map we analyzed has the similar quality as the electron density map obtained by crystallographic analysis. In fact, in the structural refinement of the OEC, we assigned the initial positions of the metal and oxygen atoms based on the highest peaks in the cryo-EM maps, and performed the structural refinement with restraints for bond distances that were taken from the initial position. This has been written in the Methods section (page 22, lines 470-473). This means that the structure was determined only from the cryo-EM data without model available. According to the reviewer's comment, we replaced the sentences "In recent years, the resolution of single-particle cryo-EM has improved to atomic resolutions without crystallization, and it has been reported that biological samples are damaged by electron beams." in the Discussion section (page 13, lines 269-277) to the following sentences.

"X-ray crystal structure analysis has been the mainstream method for determination of the three-dimensional structures of biological samples. However, the bottleneck of crystal structure analysis is crystallization because a large amount of samples having good monodispersity are required, and even if a sample having good features is obtained, crystals are not always obtained. This is particular a problem in obtaining well-diffracting crystals of membrane proteins and their complexes. In recent years, the resolution of single-particle cryo-EM structural analysis has been improved to atomic resolution^{20,21}. Due to the improved resolution, it has been reported that biological samples are damaged by electron beams²⁰."

Comment 5:

Although the manuscript deals with structure determination using cryo-EM (and SR, XFEL) techniques there have been many other methods used to get information about PS II. In particular, the status of the OEC cannot be determined without these spectroscopic techniques, see point(i). It would therefore be appropriate to mention and cite the respective references in the present work (e.g. recent reviews).

Author reply 5:

Based on the comment of the reviewer, we added a sentence "and a number of spectroscopic methods including extended X-ray absorption fine structure and electron paramagnetic resonance measurements (for ref. 9 for a recent review)", and cited a recent review article in the Introduction section (lines 47-49, page 3), to indicate that the OEC structure has also been studied by various spectroscopic methods.

Comment 6:

Although the manuscript is in general very well done, the figures should be improved. Fig. 1:

increase thickness of lines; Fig. 2: enlarge structures, maybe use stereo figures; Fig. 4: use thicker lines, enlarge (numbers are too small); Supplementary Fig. 2, 4a, b and 6: use thicker lines and colors that could be better distinguished. In the important Fig. 8 the green mesh is hard to see.

Author reply 6:

Thanks for this important comment. According to the reviewer's comment, we revised the Figs. 1, 2, 4 and Supplementary Figs. 2, 4, 6 and 8.

Responses to comments of Reviewer #4

Comment 1:

This is an interesting and important paper that explores the effects of electron beam damage on the structure of photosystem II. The work is well done and the structure of high quality. There are some unsupported statements and somewhat faulty comparisons but none are essential to the main message of the paper regarding radiation damage so can be easily revised or deleted. I recommend publication after some revisions as described below, none of which require additional experiments.

Author reply 1:

First of all, we thank the reviewer for his/her highly positive and valuable comments. According to the reviewer's comments, we modified the manuscript as follows.

Major

Comment 2:

Comparison between the 3 cryoEM datasets (page 6): Inconclusive and potentially misleading, should be removed. Since only the highest resolution dataset is discussed regarding radiation damage, the other two do not need to be included in the main text (or at all). Also the fits in Fig 1 are incorrect because the data is not linear over the fit range, yield

Author reply 2:

Thanks for this important comment. We admit that the only highest resolution data with CRYO ARM was used regarding radiation damage in the present manuscript. However, in view of the wide utilization of the Titan Krios electron microscopy in the current cryo-EM community, we think that inclusion of the Titan data would be useful for comparison with the CRYO ARM data, so would like to leave the Titan data in the current manuscript. According to the reviewer's comment, we revised Fig. 1 by deleting the points with small numbers of particles, which is now linear over the fit range.

Comment 3:

PsbY density (page 7): The authors claim PsbY was found on both PSII in the dimer, but the density is poorer. The authors need to show that this is the case *without* the application of C2 symmetry if they want to keep this statement. To support the statement that the cryoEM structure more closely represents the native state, the authors can for example perform refinement with symmetry expansion to estimate the fraction of PSII monomers containing

PsbY. Otherwise, the authors should remove this statement.

Author reply 3:

Thanks for this important comment. As shown in the following figure, even if we did the 3D classification and 3D auto-refine without C2 symmetry (subjected C1 symmetry), PsbY was still found in both of the two monomers. According to the reviewer's comment, we added the sentence "even if the 3D classification and 3D reconstruction is performed without imposing the C2 symmetry," in page 7, lines 141-142, and added the figure in the Supplementary Information (Supplementary Fig. 5).

Comment 4:

Mn-O distances (page 9): The authors point out that the Mn-O distances in the cryoEM map are consistently longer than those in the maps determined by X-ray crystallography, and list two

possible reasons. A third possible reason is the difference between the structure factors measured by X-ray crystallography, which correspond to the electron density only, vs those measured by EM, which correspond to the electrostatic potential due to nuclei and electrons together. See for example discussion regarding C-H bond lengths in Nakane et al 2020 bioRxiv. The authors need to point this out. The authors need to consider whether or not this effect affects their bond length measurements and conclusions regarding radiation damage.

Author reply 4:

Thanks for this important comment. We added a section (page 17, lines 357-365) do discuss this possibility. In Nakane et al 2020 bioRxiv, it is stated that an increase in the C-H bond length was observed due to the mismatch between the centre of electron density and the nuclear position. However, for non-hydrogen atoms, it is safe to assume that the centre of electron density and the nuclear position coincide. In fact, non-hydrogen atoms are in good agreement with the cryo-EM map. Therefore, it is considered that the mismatch between the centre of electron density and the nuclear position does not affect the Mn-O bond length, which is a non-hydrogen atom. We added a paragraph to discuss this and cited the ref. of Nakane et al. (Nature, 2020) in the revised manuscript (page 17, lines 358-366).

Comment 5:

The authors need to show some other bond distance deviations e.g. some polar bonds in the protein. The authors also need to show how was the pixel size calibrated to be 0.822, to eliminate that the deviations are not due to an overall scaling error.

Author reply 5:

Thanks for this important comment. The average bond lengths between Mg of Chl and ligand in the high-dose structure was 2.22 and the standard deviation was 0.048. According to the reviewer's comment, we added Supplementary Table 2 that listed the distances and their deviations of all Mg of Chls and their ligands. The Pixel size was determined as a value in which the map of the molecular surface and the model match well after the crystal structure was rigid body-fitted on the cryo-EM map. The value calibrated later by Au scattering was 0.823 Angstrom/pixel.

Comment 6

Resolution vs beam damage (page 11, Supplementary Fig. 7, and Methods page 19):

From the methods text it seems like the reconstructions for Supplementary Figure 7 were performed with the indicated cumulative dose, for example reconstruction (6) includes all

frames 1 - 6. Please clarify whether this is the case or not. Another map to look at the damaged structure *only* can be produced for example by summing all the frames from 6 onwards (without the early undamaged frames).

Add another panel to Supplementary Figure 7 which should show the per-frame B factor vs dose and the horizontal axis should match the horizontal axis of the resolution vs dose plot.

Author reply 6:

As the reviewer pointed out, reconstructions for Supplementary Figure 7 were performed with the indicated cumulative dose, for example reconstruction (6) includes all frames 1-6. The first one frame was specified during motion collection, and the others were summed during bayesian polishing. Since the calculation method is different, the first one frame was deleted from the original supplementary figure 7.

Since this paper focuses on how damage can be reduced, we consider that the structures with cumulative doses already showed the damage enough, and therefore do not compare it between damaged structures. According to the reviewer's comment, we deposited a plot of B-factor to supplementary figure 8 (original Supplementary Fig. 7) showing the per-frame B factor vs dose.

Comment 7:

Data availability: Please deposit the highest resolution dataset (raw movies) in EMPIAR. It is not necessary for publication but would be appreciated by the field.

Author reply 7:

According to the reviewer's comment, we deposited the highest resolution dataset (raw movies) in EMPIAR.

Comment 8:

Please add more detail to the comparisons btw. the SR XFEL and cryoEM structures in the figures. In particular, it is interesting to see if particular atoms follow specific trajectories during damage or if their positions are just randomised / blurred out.

Author reply 8:

In this study, the refinement of the high-dose and low-dose OEC structures were carried out with restraints for bond distances (Mn-O and Ca-O), and as shown in Table 3, the low-dose structure is closer to the crystal structure than the high-dose structure. However, the atomic coordinates of the high-dose and low-dose OEC do not match. Since there are no reference atoms, the trajectory of each atom cannot be traced.

Minor

Comment 9

Writing clarity needs to be improved - perhaps copy editors can assist with this.

Author reply 9:

Thank you for your comment. We have re-checked the manuscript carefully throughout the manuscript, and corrected some misses and improved the clarity.

Comment 10:

Figure 1 - if the authors decide to keep this part in the paper at all: please add more points for each particle number. Estimate errors on the slopes. Discard points at low resolution (skew the fits).

Author reply 10:

According to the reviewer's comment, we added the square of the correlation coefficient (R^2) and deleted the points at low resolution in Figure 1. However, each point for each particle number was taken randomly, and we consider that these points reflect the real resolution already. Taken more random points for each particle number may give some deviations for each resolution, but they should not be deviated significantly from the one that was shown. Thus, we did not add more points for each particle number.

Comment 11:

Line 124, page 6 - 'size' should read 'number'

Author reply 11:

We revised it as suggested.

Comment 12:

Overall structure (p.7): Please comment whether or not all pigments, as found in the X-ray structure, are present in the cryoEM map. This is not clear from the supplementary figures.

Author reply 12:

Yes, all pigments found in the X-ray structure are present in the cryo-EM structure. We added the following sentence in Overall structure of PSII section (lines 135-136, page 7) according to the reviewer's comment.

“...and all pigments found in the X-ray structure are present in the cryo-EM map.”

Comment 13:

Line 272, page 13 - symbol before 3.7 not displayed.

Author reply 13:

We deleted it as suggested.

Comment 14:

Discrepancy between methods for cryoEM data collection (page 17) and Table 1 (page 33): Methods say Falcon 3 linear mode, table says counting mode. The flux seems consistent with linear mode. The dose rate, as stated in the text, for all 3 experiments, does not equal the total dose divided by the exposure time as stated in the table. Please correct all of these.

Author reply 14:

Thanks for this important comment. The table was correct, and we corrected the Methods section to indicate Falcon 3 in electron counting mode. We also modified the dose rates in the text to match those in the Table.

Comment 15:

CryoEM image processing (page 17 - 18): repetitive text for the 3 datasets can be summarised in Table 1 instead.

Please state which program/method was used for particle picking.

Author reply 15:

Three datasets were processed using RELION3.0 in the almost same procedure. To make the references for the auto-picking, about 2,000 particles were manually picked from corrected micrographs and supplied to low pass filter of 20 Å, and then subjected to reference-free 2D classification. Auto-picking was performed with RELION3.0 based on the template-matching using the good classes as the reference. We added these sentences in the Methods section as suggested.

Comment 16:

Supplementary Figs 1 & 3. Please state how the micrograph contrast was adjusted, low pass filtering, etc.

Author reply 16:

Thanks for this important comment. As described in the Methods section, movie frames were aligned and summed using the MotionCor2 software to obtain a final dose weighted image. Estimation of the contrast transfer function (CTF) was performed using the CTFFIND4 program. As described in reply 15, we supplied low pass filter of 20 Å during the manual picking. We added these programs in Supplementary Figs. 1 and 3 as suggested.

REVIEWERS' COMMENTS:

Reviewer #1 (Remarks to the Author):

The revised version from Kato et al. has improved by carefully implementing suggestions from the referees. The discussion of the Mn-O bond length has been broadened as high and low dosage has been compared to SR and XFEL data.

Minor points.

In the added text lines 281-287 the word obtained is excessively used(283). The claim that the subunit PsbY can with cryo-EM resolved as indication for a more native state compared to the absence or partial absence in crystal structures with SR or XFEL is far fetched or speculative.

Reviewer #2 (Remarks to the Author):

Kato et al. have carefully addressed reviewers' comments and concerns in their revised manuscript on "High-resolution cryo-EM structure of photosystem II: Effects of electron beam damage". A few issues need to be addressed before it should be accepted for publication.

Logic in lines 220-225/lines 253-257 is invalid. It is an oxidation reaction (removal of an electron) to convert YD to YD⁺ radical. Cryo-EM should reduce YD⁺ radical to the native YD. It is possible that SR and XFEL structures contain partially oxidized YD⁺ radical that was present in the dark-adapted samples, which is why the water molecule is disordered between two alternative positions. In cryo-EM maps, it is fully reduced with one single position for the water molecule. There is an unnecessary repetition here as well.

Arguments in lines 339-351 are not very convincing and are very wordy. It is true that the majority of metalloenzyme are reduced as a consequence of X-ray irradiation. X-ray irradiation splits water molecules and protein molecules into highly oxidizing hydroxyl radical, highly reducing H radical and hydrated electrons. Oxidizing hydroxyls are inserted into the protein structures while reducing equivalents follow intrinsic electron transfer pathways to the metal cluster. PSII is not an every metalloenzyme because it has two separated pathways, one for electron transfer to quinone site, and the other for oxidizing equivalent transfer to the OEC. The major damaging species from electron irradiation is hydrated electrons. This section requires an extensive editing. Please remove unnecessary repetition. I would start this discussion section something like:

It has been reported that about 80% Mn ions of OEC is reduced to divalent cations at a dose of 5 MGy. If this dose-reduction relationship is applied to cryo-EM, which has not yet been validated, the low dose of 12.2 MGy used in this study would reduce 90% Mn ions of the OEC to Mn²⁺ ion.

I would not repeat an interpretation or a speculation that is based on an unproved extrapolation.

There are two results sections: "Electron beam damage to the PSII structure" and "PSII structure at reduced electron beam dosages". Description of metal-metal distances of the OEC is very difficult to read and to compare, particularly in different sections. Authors should consider merging them into

one section and simplify the description. An example of simpler description is as follows. The Mn-Mn distances of the OEC determined from the cryo-EM map are 2.8 Å, 3.5 Å, 5.0 Å, 3.1 Å, 5.4 Å, and 2.7 Å for Mn1-Mn2, Mn1-Mn3, Mn1-Mn4, Mn2-Mn3, Mn2-Mn4, and Mn3-Mn4, respectively. Corresponding distances in the SR structure are xx Å (increased in cryo-EM structure by +yy Å), xx Å (+yy Å), xx Å (+yy Å), etc; and corresponding distance in the XFEL structure are xx Å (increased by +yy Å), xx Å (+yy Å), xx Å (+yy Å), etc. This makes description clear before the reader divides into a very large complicated table 3.

Line 94: “are not only” \ “not only are”

Line 119: “the major one causing the resolution difference here could be the electron beam source used” \ “the major factor affecting the resolution could be different quality of electron beams from different instruments”

Line 128: Å² unit should be added after each numerical value.

Line 244: “restored” \ “retained”

Line 306: “This is considered to be partially resulted from the successful dose-weighted correction in the Bayesian polishing step.” \ This likely resulted from a successful dose-weighted correction in the Bayesian polishing step.

Lines 318-319: This structure was compared with that obtained with the high-dose, and it was found that the disulfide bond in the PsbO was recovered, and A344 of the D1 subunit was returned to the single conformation similar to the SR and XFEL structures. \ When the low-dose and high-dose cryo-EM structures were compared, it was found that a disulfide bond in PsbO was retained and A344 of the D1 subunit remained in the single conformation as found in the SR and XFEL structures.

Line 340: “According to this” \ “According to this dose”

Lines 362-365: The sentences “The second cause ... this may be caused by...” read very poorly. I would start something like: Some discrepancy between the cryo-EM and XFEL structures is due to coordinate errors associated with model refinement and experimental errors. In the absence of spatial resolution, distribution of coordinate errors may be different between cryo-EM and XFEL structures. I would reduce the Discussion section by at least two thirds.

Table 3: All numbers should have consistent significant digitals (two decimal digitals).

Reviewer #3 (Remarks to the Author):

The authors have carefully revised their manuscript and have satisfactorily answered all my questions. The manuscript can now be published.

Minor points

line 50: ..(see ref. 9 for a recent review). The

line 84/85: ... decarboxylation of amino acids (better: amino acid residues) and chemical reduction
....

Please check the correct use of articles in all the text, in particular in the added parts of the manuscript.

Responses to the comments of Reviewer #1

Remarks to the Author:

Comment 1:

The revised version from Kato et al. has improved by carefully implementing suggestions from the referees. The discussion of the Mn-O bond length has been broadened as high and low dosage has been compared to SR and XFEL data.

Author reply 1:

First of all, we thank the reviewer for his/her highly positive and valuable comments. According to the reviewer's comments, we modified the manuscript as follows.

Minor points.

Comment 2:

In the added text lines 281-287 the word obtained is excessively used(283). The claim that the subunit PsbY can with cryo-EM resolved as indication for a more native state compared to the absence or partial absence in crystal structures with SR or XFEL is far fetched or speculative.

Author reply 2:

According to the reviewer's comment, we completely removed the original sentence "...even if a sample having good features is obtained, crystals are not always obtained.", and modified the original sentence regarding PsbY into the following one (page 14, lines 284-285):

"This indicates that the structure solved by cryo-EM may represent the physiological state more closely".

Responses to comments of Reviewer #2

Remarks to the Author:

Comment 1:

Kato et al. have carefully addressed reviewers' comments and concerns in their revised manuscript on "High-resolution cryo-EM structure of photosystem II: Effects of electron beam damage". A few issues need to be addressed before it should be accepted for publication.

Author reply 1:

First of all, we thank the reviewer for his/her highly positive and valuable comments. According to the reviewer's comments, we modified the manuscript as follows.

Comment 2:

Logic in lines 220-225/lines 253-257 is invalid. It is an oxidation reaction (removal of an electron) to convert YD to YD⁺ radical. Cryo-EM should reduce YD⁺ radical to the native YD. It is possible that SR and XFEL structures contain partially oxidized YD⁺ radical that was present in the dark-adapted samples, which is why the water molecule is disordered between two alternative positions. In cryo-EM maps, it is fully reduced with one single position for the water molecule. There is an unnecessary repetition here as well.

Author reply 2:

Thanks for this important comment. According to the reviewer's comment, we replaced the sentence "This may reflect the again reflect the electron beam induced damage, which causes reduction of Y_D and breaks the hydrogen-bond to Y_D⁺." by the following one, and we deleted the sentence "However, similar to the high-dose structure, the water molecule near Y_D are connected to D2-Arg180 in an ordered manner and not hydrogen-bonded to Y_D (Supplementary Fig. 7), indicating that some electron beam damage remained." in lines 254-257.

"This may reflect the electron beam induced damage, which causes full reduction of Y_D⁺ and breaks the hydrogen-bond to Y_D (see Methods section)."

Comment 3:

Arguments in lines 339-351 are not very convincing and are very wordy. It is true that the majority of metalloenzyme are reduced as a consequence of X-ray irradiation. X-ray irradiation splits water molecules and protein molecules into highly oxidizing hydroxyl radical, highly reducing H radical and hydrated electrons. Oxidizing hydroxyls are inserted into the protein

structures while reducing equivalents follow intrinsic electron transfer pathways to the metal cluster. PSII is not an every metalloenzyme because it has two separated pathways, one for electron transfer to quinone site, and the other for oxidizing equivalent transfer to the OEC. The major damaging species from electron irradiation is hydrated electrons. This section requires an extensive editing. Please remove unnecessary repetition. I would start this discussion section something like:

It has been reported that about 80% Mn ions of OEC is reduced to divalent cations at a dose of 5 MGy. If this dose-reduction relationship is applied to cryo-EM, which has not yet been validated, the low dose of 12.2 MGy used in this study would reduce 90% Mn ions of the OEC to Mn²⁺ ion.

I would not repeat an interpretation or a speculation that is based on an unproved extrapolation.

Author reply 3:

Thanks for this important comment. According to the reviewer's comment, we replaced the sentences to the following one in page 16, lines 329-332.

“It has been reported that about 80% Mn ions of OEC is reduced to divalent cations at a dose of 5 MGy. If this dose-reduction relationship is applied to cryo-EM, which has not yet been validated, the low dose of 12.2 MGy used in this study would reduce 90% Mn ions of the OEC to Mn²⁺ ion.”

We also modified and condensed the whole section to avoid unnecessary repetitions here.

Comment 4:

There are two results sections: “Electron beam damage to the PSII structure” and “PSII structure at reduced electron beam dosages”. Description of metal-metal distances of the OEC is very difficult to read and to compare, particularly in different sections. Authors should consider merging them into one section and simplify the description. An example of simpler description is as follows.

The Mn-Mn distances of the OEC determined from the cryo-EM map are 2.8 Å, 3.5 Å, 5.0 Å, 3.1 Å, 5.4 Å, and 2.7 Å for Mn1-Mn2, Mn1-Mn3, Mn1-Mn4, Mn2-Mn3, Mn2-Mn4, and Mn3-Mn4, respectively. Corresponding distances in the SR structure are xx Å (increased in cryo-EM structure by +yy Å), xx Å (+yy Å), xx Å (+yy Å), etc; and corresponding distance in the XFEL structure are xx Å (increased by +yy Å), xx Å (+yy Å), xx Å (+yy Å), etc. This makes description clear before the reader divides into a very large complicated table 3.

Author reply 4:

Thanks for this important comment. We modified the original text based on the reviewer's comments (lines 186-192, 201-204, lines 254-255, lines 261-263, revised manuscript).

Comment 5:

Line 94: "are not only", "not only are"

Author reply 5:

We revised it as suggested.

Comment 6:

Line 119: "the major one causing the resolution difference here could be the electron beam source used", "the major factor affecting the resolution could be different quality of electron beams from different instruments"

Author reply 6:

We modified it as suggested; thank you.

Comment 7

Line 128: Å² unit should be added after each numerical value.

Author reply 7:

We added the Å² unit to each numerical value.

Comment 8:

Line 244: "restored", "retained"

Author reply 8:

We revised it as suggested.

Comment 9:

Line 306: "This is considered to be partially resulted from the successful dose-weighted correction in the Bayesian polishing step.", "This likely resulted from a successful dose-weighted correction in the Bayesian polishing step."

Author reply 9

We revised it as suggested.

Comment 10:

Lines 318-319: “This structure was compared with that obtained with the high-dose, and it was found that the disulfide bond in the PsbO was recovered, and A344 of the D1 subunit was returned to the single conformation similar to the SR and XFEL structures.”, “When the low-dose and high-dose cryo-EM structures were compared, it was found that a disulfide bond in PsbO was retained and A344 of the D1 subunit remained in the single conformation as found in the SR and XFEL structures.”

Author reply 10:

We revised it as suggested; thank you.

Comment 11:

Line 340: “According to this”, “According to this dose”

Author reply 11:

According to the previous comments of you, we replaced this sentence by the following one.

“If this dose-reduction relationship is applied to cryo-EM, which has not yet been validated, the low dose of 12.2 MGy used in this study would reduce 90% Mn ions of the OEC to Mn²⁺ ion.”

Comment 12:

Lines 362-365: The sentences “The second cause ... this may be caused by...” read very poorly. I would start something like: Some discrepancy between the cryo-EM and XFEL structures is due to coordinate errors associated with model refinement and experimental errors. In the absence of spatial resolution, distribution of coordinate errors may be different between cryo-EM and XFEL structures. I would reduce the Discussion section by at least two thirds.

Author reply 12:

We revised it as suggested. We also reduced the Discussion section to some extent. However, due to the importance of the discussion section, a reduction of 1/3 was not possible.

Comment 13:

Table 3: All numbers should have consistent significant digitals (two decimal digitals).

Author reply 13:

We revised it as suggested.

Responses to comments of Reviewer #3

Comment 1:

Remarks to the Author:

The authors have carefully revised their manuscript and have satisfactorily answered all my questions. The manuscript can now be published.

Author reply 1:

First of all, we thank the reviewer for his/her highly positive and encouraging comments. According to the reviewer's comments, we modified the manuscript as follows.

Comment 2:

Minor points

line 50: ..(see ref. 9 for a recent review). The

Author reply 2:

We removed it; thank you.

Comment 3:

line 84/85: ... decarboxylation of amino acids (better: amino acid residues) and chemical reduction

Please check the correct use of articles in all the text, in particular in the added parts of the manuscript.

Author reply 3:

We revised it as suggested, and checked all parts of the text to make it sure there are no errors existed.